# epiAneufinder identifies copy number alterations from single-cell ATAC-seq data

Akshaya Ramakrishnan [1,4], Aikaterini Symeonidi [1,2,4] ✉, Patrick Hanel[1,2], Katharina T. Schmid [2], Maria L. Richter [2], Michael Schubert [3] & Maria Colomé-Tatché[1,2] ✉

Single-cell open chromatin profiling via scATAC-seq has become a mainstream measurement of open chromatin in single-cells. Here we present epiAneu-finder, an algorithm that exploits the read count information from scATAC-seq data to extract genome-wide copy number alterations (CNAs) for individual cells, allowing the study of CNA heterogeneity present in a sample at the single-cell level. Using different cancer scATAC-seq datasets, we show that epiA-neufinder can identify intratumor clonal heterogeneity in populations of single cells based on their CNA profiles. We demonstrate that these profiles are concordant with the ones inferred from single-cell whole genome sequencing data for the same samples. EpiAneufinder allows the inference of single-cell CNA information from scATAC-seq data, without the need of additional experiments, unlocking a layer of genomic variation which is otherwise unexplored.

Aneuploidy and copy number alterations describe DNA duplication and deletion events that range from a small number of base pairs in the genome to whole chromosomes. Both conditions have been involved in disease, and are especially prominent in cancer[1]. In fact, more than ~90% of solid tumors are aneuploid[2], leading to the hypothesis that aneuploidy confers a growth advantage to cancer cells[3]. However, the relationship between aneuploidy and tumorigenesis remains not clearly understood[4,5]. Meanwhile, aneuploidy is highly detrimental to normal cell development and growth[6,7].

Because of its biological and clinical relevance, aneuploidy is widely studied using different experimental strategies[8]. Traditionally, CNAs have been studied with spectral karyotyping or (interphase) fluorescence in situ hybridization (FISH), which are single-cell low-throughput methods that lack precise genomic resolution[8]. On the contrary, comparative genomic hybridization (CGH), whole exome sequencing (WES) or whole-genome sequencing (WGS) provide high genomic resolution but do so at the expense of losing single-cell resolution[8]. Single-cell whole-genome sequencing (scWGS) offers a compromise, interrogating copy number gains and losses in a high throughput fashion at the single-cell level and at a higher genomic resolution than FISH or spectral karyotyping[8–10]. Despite being considered the gold-standard ground truth for the quantification of CNA heterogeneity in large single-cell populations[9], scWGS is not often used in the laboratory compared to other single-cell sequencing techniques. Attempts have been made to call copy number variations from single-cell gene expression data, however, at high genomic resolution these are confounded by physiological variation in expression levels[11], and calling CNAs from single-cell gene expression alone proves challenging[12–16].

A newer single-cell measurement technique, the single-cell Assay for Transposase-Accessible Chromatin using sequencing (scATAC-seq), has become a mainstream measurement in single cells[17], partially owing to its implementation in the 10x platform[18]. Single-cell chromatin openness measurements require the sequencing of the DNA of the cell, instead of the RNA like in scRNA-seq, hence scATAC-seq measurements have the potential to better recapitulate the DNA content from single cells. However, scATAC-seq data is extremely sparse, making it challenging to directly extract copy number calls from the data[18,19].

[1]Institute of Computational Biology, Helmholtz Zentrum München, German Research Center for Environmental Health, Neuherberg, Germany. [2]Biomedical Center (BMC), Physiological Chemistry, Faculty of Medicine, LMU Munich, Planegg-Martinsried, Germany. [3]Oncode Institute, Division of Cell Biology, Netherlands Cancer Institute, Plesmanlaan 121, 1066 CX Amsterdam, the Netherlands. [4]These authors contributed equally: Akshaya Ramakrishnan, Aikaterini Symeonidi. ✉e-mail: aikaterini.symeonidi@helmholtz-munich.de; maria.colome@bmc.med.lmu.de

In this paper, we present an algorithm, epiAneufinder, that calls copy number alterations at the single-cell level from scATAC-seq data. EpiAneufinder uses binary segmentation combined with an appropriate choice of distance measure to identify putative breakpoints in the genome, and subsequently calls gains and losses per identified segment. EpiAneufinder can identify single-cell CNAs from scATAC-seq data alone, without the need of a reference euploid sample and without the need to supplement the data with other data modalities. We demonstrate the performance of epiAneufinder by applying it to different scATAC-seq cancer datasets for which an orthogonal measurement of CNAs is available (either scWGS or WGS). In conclusion, epiAneufinder allows the addition of an extra level of genetic information, namely CNAs, to scATAC-seq or single-cell multi-ome (combined scRNA and scATAC-seq) data, without the need of any extra experimental effort. EpiAneufinder is available as an R package at https://github.com/colomemaria/epiAneufinder.

## Results

### epiAneufinder algorithm

The goal of epiAneufinder is to segment the genome into regions of gain, loss, and normal copy number per single cell. To do that, epiAneufinder uses the number of reads from scATAC-seq data mapping to a genomic region as a proxy of the number of DNA copies present in that region, for every single cell. To overcome the coverage sparsity inherent to single-cell sequencing, lowly covered cells are filtered out, the genome is binned into equally sized windows (by default, window size is 100,000 bp) and the number of mapped reads per window is quantified (Fig. 1a). Furthermore, we remove the ENCODE blacklisted set of regions[20], composed of certain genomic locations, such as telomeric ends and repetitive regions, that have systematic biases in their mappability and that would therefore bias the copy number inference. For every dataset, epiAneufinder also removes bins that have zero counts in >85% of all the cells, to discard genomic areas that have low mappability in every dataset specifically. All parameters can be adjusted by the user.

After that, the binned dataset is GC corrected using a LOESS fit. The correction factor is obtained by fitting the raw read counts to the GC content per bin:

$$x_t^{GC} = x_t \cdot f_{GC} = x_t \cdot \frac{mean(x_t)}{Loess(x \sim GC)_t}, \tag{1}$$

where $x_t$ is the number of reads in bin $t$, $mean(x_t)$ is the average read count per bin, and $GC$ is the percentage of base pairs which are GC per bin.

After data preparation, epiAneufinder applies a binary segmentation algorithm to each single cell separately. Binary segmentation is a technique used to detect change points in signals by identifying positions of data distribution changes. In the case of CNA calling, the assumption is that the distribution of reads mapping per bin is different for a gained, a lost, and a normal copy number region (Supplementary Fig. 1). The goal is to identify the positions in the genome where the distribution changes take place. To do that, the algorithm scans the genome of every single cell per chromosome, and calculates the Anderson–Darling (AD) distance ($d_{AD}$) between the read distributions at the left and right of every bin (Fig. 1b). The AD test is a nonparametric test that measures agreement between distributions. It was initially developed to check for normality[21], but it was subsequently modified[22] to measure the distance between any two empirical distributions:

$$d_{AD} = A_{nm}^2 = \frac{nm}{N} \int_{-\infty}^{\infty} \frac{\{F_n(x) - G_m(x)\}^2}{H_N(x)\{1 - H_N(x)\}} dH_N(x), \tag{2}$$

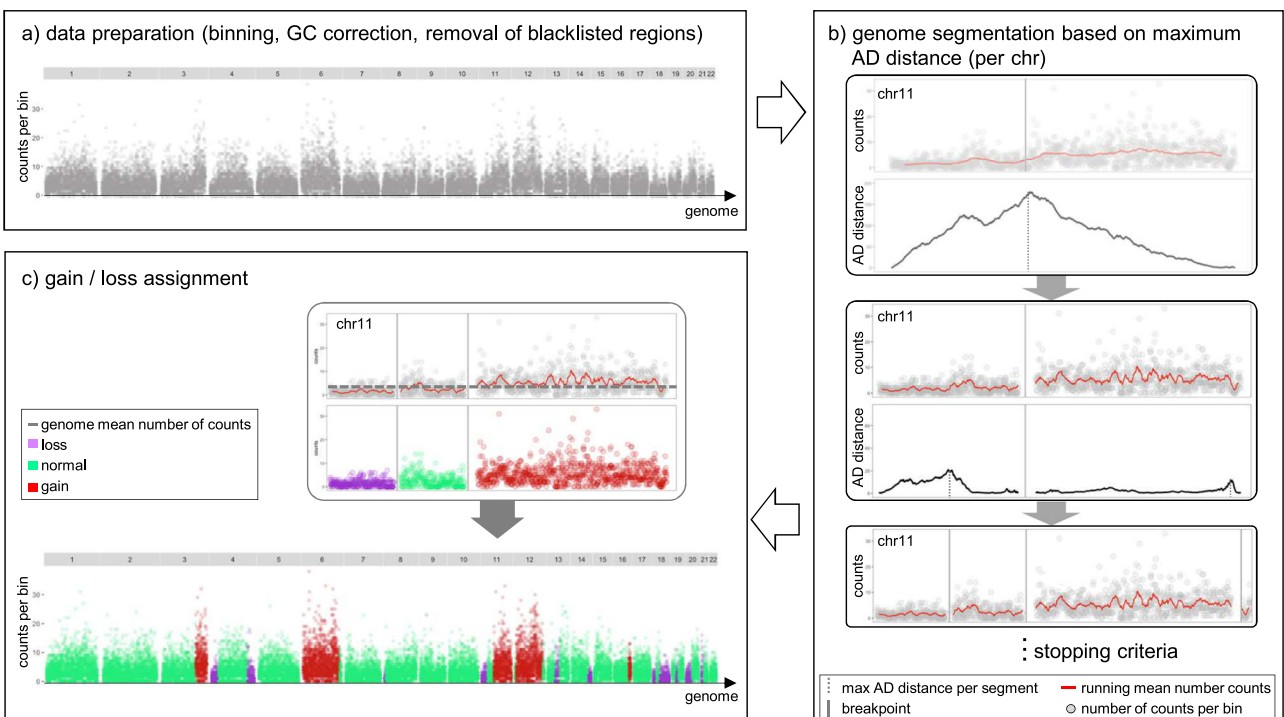

**Fig. 1 | epiAneufinder algorithm. a** The genome is binned (default 100,000-bp windows) and the number of reads in every bin are quantified. Blacklisted regions are removed, and the data is GC corrected. Bins with zero counts in >85% of all the cells are removed. **b** Binary segmentation is applied iteratively per chromosome by computing the AD distance between the left and right count distribution between bins. In every segment, the position with the highest AD distance is considered a breakpoint until a stopping criteria is reached. **c** After breakpoints have been identified per chromosome, every segment is assigned to the state loss, normal or gain based on the read count fold change over the genome-wide mean.

where $F_n(x)$ and $G_m(x)$ are the read distribution functions for the two genomic segments to compare (with lengths $n$ and $m$, respectively), and $H_N(x)$ is the distribution function of the combined segments: $H_N(x) = \{nF_n(x) + mG_m(x)\}/N$, with $N = n + m$. It was chosen here to detect copy number differences because it emphasizes the difference between distribution tails.

The position in every chromosome that maximizes the AD distance is kept as the most likely breakpoint (Fig. 1b). The same procedure is then repeated iteratively on the two resulting segments, until a total number of breakpoints are identified per chromosome (by default 15) (Fig. 1b). After all the breakpoints have been identified genome-wide, epiAneufinder prunes out breakpoints with an AD distance lower than the genome-wide mean, to remove low AD distance breakpoints. This pruning procedure assumes that less than 15 breakpoints are present per chromosome. In situations where more breakpoints are expected, the parameter for the upper number of breakpoints can be modified by the user.

Finally, every segment is assigned to the state gain, loss, or normal copy number. To do that, epiAneufinder calculates the trimmed-average number of reads per identified segment, defined as the average number of reads per bin in every segment without the lowest 0.1 and highest 0.1 quantile bins (these values can be adjusted by the user). The segments with a trimmed-average number of reads with z-score between $[-1, 1]$ are assigned to the genome-wide mean. Then, for every segment, the algorithm calculates its rounded integer fold change over the genome-wide average number of reads. For a value of 0 the segment is assigned to the state "loss", for a value of 1 to the state "normal" and for a value $>= 2$ to the state "gain" (Fig. 1c). Precise quantification of copy numbers beyond "gains" and "losses" is not possible due to the sparsity of scATAC-seq data; and full genome duplications and deletions cannot be identified, because they change the genome-wide mean openness value. For visual representation and to identify CNA clones, the single cells are clustered based on their copy number profiles using Euclidean distance and Ward Clustering.

In summary, epiAneufinder takes scATAC-seq BAM files or 10x fragment files as input, and outputs a RDS file and a TSV file with the identified copy number states for each bin per cell, labeled as "loss" (0), "normal" (1) or "gain" (2). Other intermediate result files are also provided, such as the (GC corrected) binned number of reads per cell (RDS file), and the identified breakpoints per cell and per chromosome with their associated AD distance (RDS file). Moreover, plotting functions are available via epiAneufinder, to plot the resulting single-cell karyotypes and the clustering results. Run times are available in Supplementary Table S1.

## epiAneufinder CNAs are concordant with the ones obtained from (sc)WGS

To demonstrate the performance of epiAneufinder, we analyzed several scATAC-seq datasets. First, we analyzed a recent scATAC-seq dataset for a gastric cancer cell line. This dataset contains ~3500 aneuploid single-cells from the gastric adenocarcinoma cell line SNU601 with a mean coverage of 75,013 fragments per cell. The SNU601 cell line is known to contain a complex subclonal structure with multiple clones that harbor different copy number calls[23]. The same cell line was profiled using scWGS[23], a measurement which can be used as an independent ground truth for comparing the identified scATAC-seq CNA calls to.

We segmented the SNU601 genome into 100,000-bp windows and we quantified the number of reads per cell in every window. EpiAneufinder was applied to every cell in the population with standard parameters, breakpoints were identified, and every segment was assigned to the state "gain", "loss" or "normal" (Supplementary Fig. 2a). The smallest CNA identified was 100 kb (1 bin), the largest gain was 266,700 kb and the largest loss was 361,900 kb (Supplementary Fig. 2b). Varying the total number of breakpoints called per

chromosome did not substantially change the CNA results (Supplementary Fig. 2c, d and Supplementary Table 1). We identified several karyotype clusters in the dataset (Fig. 2a and Supplementary Fig. 2a). Nearly all cells presented a gain in chromosome 19, and the main source of variation was the absence (cluster 1) or presence (all other clusters) of CNAs in the remaining part of the genome. Cluster 2 was further split into two groups, mainly differentiated by the presence or absence of a whole chromosome gain in chromosome 11. Cluster 1 was found to be nearly disomic. Cluster 3 could be split into different minor groups, the major source of variation being the presence of a longer or shorter gain in chromosomes 3 and 6, as well as the presence of losses in chromosome 18.

To validate our results, we analyzed a scWGS dataset for the same cell line, with 1531 cells (average coverage of 707,188 reads per cell)[23]. Copy number gains and losses were called using the package aneufinder, designed to work with single-cell DNA sequencing data[24], using the same 100,000-bp window size as for the scATAC-seq. Despite the fact that the two datasets were produced by different laboratories using different techniques, the scWGS data presented a very similar CNA profile as the scATAC-seq dataset (Fig. 2b and Supplementary Fig. 3a).

To quantify the similarities between the two copy number profiles, we constructed the in-silico pseudo-bulk copy number profiles, calculated as the mean of the gains and losses in every bin in the population. In general, the same pseudo-bulk gain and loss profile was observed for both modalities (Fig. 2c), and the pseudo-bulk DNA-seq and ATAC-seq profiles were highly similar, with a maximum F1 score for the loss, gain and normal states of 0.88, 0.93, and 0.85, respectively (Supplementary Fig. 3b, c). Considering the DNA-seq profile as the observed (true) value, and the scATAC-seq profile as the predicted one, the mean square error (MSE) of the detected scATAC-seq CNAs was computed, which was as low as 0.09 genome-wide (Supplementary Table S2).

Generally, the pseudo-bulk scATAC-seq copy number profile tended to be less penetrant than the scWGS one, an effect that was observed both for gains and for losses (Fig. 2c). The scWGS dataset presented some sharp singularities around the centromeres/pericentromeres that were not detected on the scATAC-seq dataset; while a very penetrant gain on chromosome 5 observed on the scWGS dataset was not recovered in the scATAC-seq dataset (Fig. 2c). These discrepancies could be due to the stark differences in experimental protocols between scWGS and scATAC-seq measurements, combined with the fact that the DNA and ATAC experiments were performed independently by different laboratories (Wu et al.[19] and Andor et al.[23]), and it has been documented that despite its assumed homogeneity, cell lines can be highly variable[25].

Apart from the SNU601 cell line, we used epiAneufinder to analyze scATAC-seq data for two further aneuploid cell lines (HCT116[26] and colo320[27]), as well as for a primary glioblastoma sample[28]. As a ground truth dataset, scWGS data was also available for the HCT116 cell line[29], while WGS was available for the colo320 cell line[30] and for the primary glioblastoma[31]. In all three cases, epiAneufinder CNAs were highly similar to the ones called using DNA information (epiAneufinder was used with standard parameters and comparison to the whole-genome sequencing results was done using pseudo-bulk aggregates) (Supplementary Figs. 4–6 and Supplementary Tables 2 and 3).

Finally, epiAneufinder was also applied to multiple scATAC-seq datasets from diploid samples, consisting of seven human brain samples[32] (four scATAC-seq datasets and three multi-ome datasets), one PBMC sample, and one bone marrow sample[18]. Comparing the epiAneufinder results (using standard parameters) to an euploid baseline, we found minimal deviations from diploidy, showing that epiAneufinder recognizes the absence of CNVs and correctly identifies euploid samples (Supplementary Fig. 7 and Supplementary Table 4).

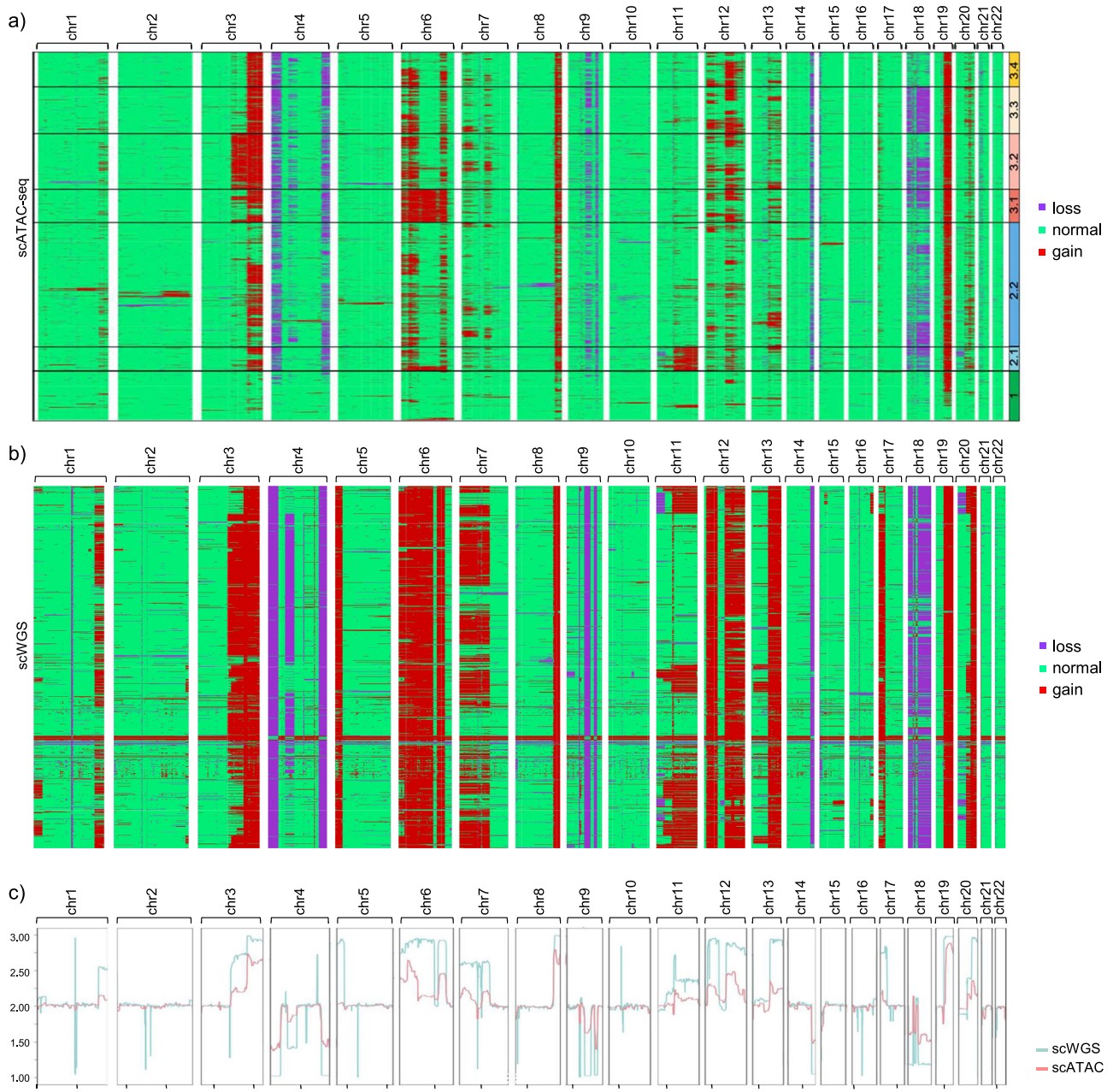

**Fig. 2 | SNU601 cell line copy number variations. a** CNAs for the SNU601 cell line obtained from scATAC-seq data. Every row is a cell, and every column is a chromosome. Karyotype clones are indicated in black boxes and numbered. **b** CNAs for the SNU601 cell line obtained from scWGS data. Every row is a cell, and every column is a chromosome. **c** Pseudo-bulk CNA profiles for the DNA and ATAC data (*x* axis shows the bin number).

## epiAneufinder outperforms other single-cell CNA calling methods

Multiple algorithms have been developed to call CNAs from scRNA-seq data, and one algorithm has previously been developed that calls CNAs from scATAC-seq data (Copy-scAT)[28]. We have compared the results from these algorithms to the ones obtained by epiAneufinder in datasets that contain multiple data modalities: the SNU601 cell line (scATAC-seq[19], scRNA-seq and scWGS[23]), the HCT116 cell line (multivariate scATAC-seq and scRNA-seq[26] and scWGS[29]) and the colo320 cell line (multi-ome data (10x)[27] and bulk WGS[30]). For calling CNAs from scRNA-seq, three methods were used: InferCNV[16], CaSpER[12] and copyKat[14]. A reference euploid dataset[33–35] was required in all RNA methods to normalize the data.

We evaluated the correlation between the genome-wide results for each RNA method compared to the scWGS or WGS results using pseudo-bulk aggregates by mapping the gene-based CNA results to the genomic bins of the (sc)WGS and epiAneufinder results. Only bins where results were available for all methods were included in the comparison.

For the SNU601 cell line, the correlation between the scWGS CNAs and the CNAs obtained by all the different methods and modalities was relatively high (Fig. 3a). Both scATAC-seq-based methods (epiAneufinder & Copy-scAT) performed better than scRNA-seq-based methods: the best scRNA-seq method, InferCNV, reached a correlation of 0.61, compared to epiAneufinder (the best scATAC-seq method) with a correlation of 0.86. In contrast, all scRNA-seq-based methods

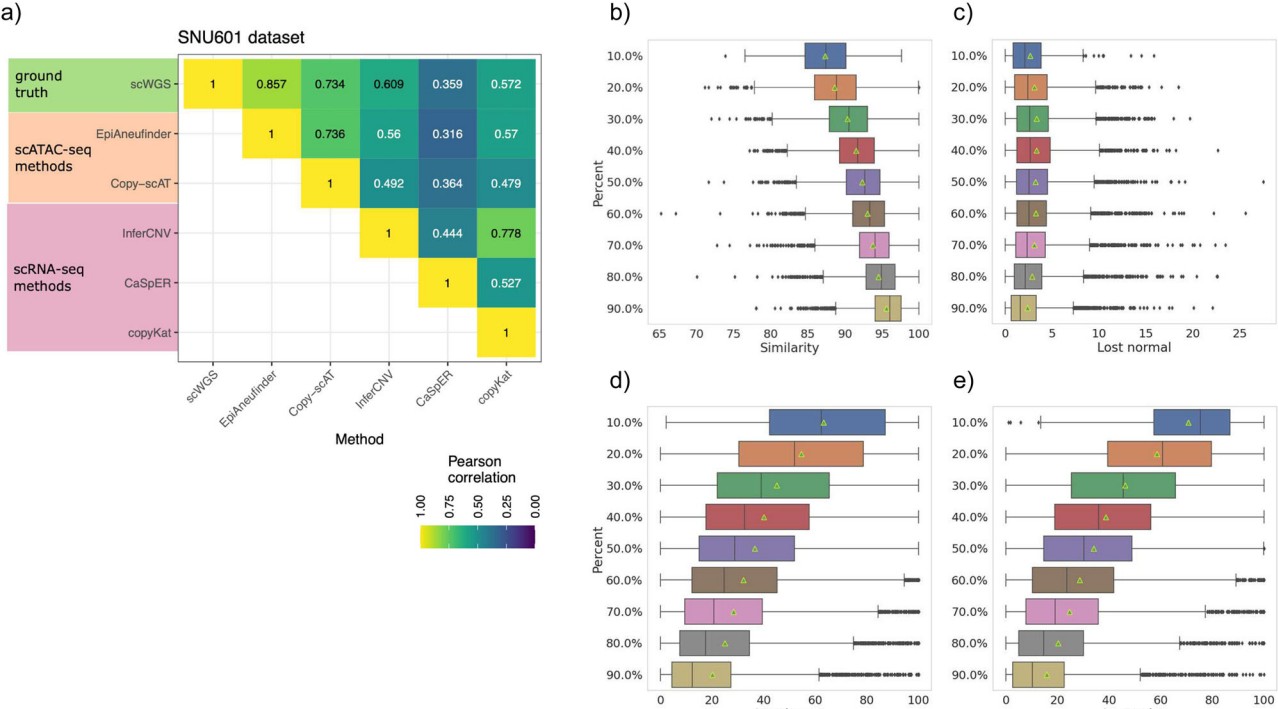

**Fig. 3 | Performance evaluation. a** Pearson correlation between pseudo-bulk profiles for all the methods used to call CNAs in the SNU601 dataset. **b** Percent of genome which is unchanged upon downsampling. **c** Percent of normal calls, **d** loss calls, and **e** gain calls which are lost upon downsampling. The number of cells in each sub-sampling percentage are in Supplementary Table S6. The boxplots represent with the vertical line in the middle of the plot the median value and the triangle the mean value. Lower and upper boundaries represent the first and third quartile, respectively. Largest and smallest observed values are shown (whisker lines) and the asterisks denote the outliers.

performed worse on the other two datasets (HCT116 and Colo320 cell lines) (Supplementary Fig. 8), with correlations in the range of 0.2–0.4 when compared to the DNA-based results. The problems to estimate CNAs based on scRNA-seq data for all the presented datasets could also be caused by the choice of a suitable reference sample, which especially for cell lines is very difficult to obtain.

For the CNAs obtained from scATAC-seq data, epiAneufinder outperformed Copy-scAT (Supplementary Fig. 8). We quantified more carefully the results of epiAneufinder and Copy-scAT for the SNU601 cell line. In contrast to the comparison with the scRNA-seq methods, a far larger part of the genome could be included in the comparison. Copy-scAT, per default, only provides chromosome-arm resolution copy numbers, visible in the very coarse-grain profile in the line plot. Nevertheless, the estimated CNAs between Copy-scAT and epiAneufinder agreed well overall (correlation of 0.76). However, epiAneufinder correlated better than Copy-scAT with the WGS results (0.86 vs 0.74).

Overall, the comparison of epiAneufinder to methods for calling CNAs from scRNA-seq data as well as one method for calling CNAs from scATAC-seq data shows that epiAneufinder more accurately reflects the WGS results.

### CNA identification is affected by sequencing read depth
To assess the robustness of epiAneufinder's gain and loss calls to sequencing depth, we performed a simulation study. The SNU601 cell line dataset[19] was downsampled from 100% coverage to only 10% of the initial reads (Supplementary Tables 5 and 6), and we used epiAneufinder with standard parameters to identify gains and losses in the population (Supplementary Fig. 9a). For every downsampled dataset, cells and bins that did not comply with the quality controls of epiAneufinder were removed (Supplementary Table 6). The results of this simulation showed that, even when only 10% of the original number of

reads were retained, on average over all the cells that passed quality control, 87% of the total genome remained in the same copy number state as identified in the fully covered dataset (Fig. 3b). Since the majority of the genome was in the normal state, we also investigated the robustness of copy number losses and gains calls separately upon downsampling. Percentage-wise, the identification of copy number losses was less affected by the downsampling than the identification of copy number gains: when considering only 50% of the initial coverage, ~28% of bins with a copy number loss in the fully covered dataset were not identified as a loss any more (lost losses), compared to ~30% of gains that were not identified any longer (lost gains) (Fig. 3d, e). As expected, the normal calls were the least affected (only 2.1% of the normal bins changed state at 10% downsampling) (Fig. 3c). For the lowest downsampling point at 10% however, the data became very sparse and less than 15% of the cells passed quality control.

Considering the fully covered dataset as the ground truth, we computed precision, recall, and F1 scores for every state (Supplementary Fig. 9b and Supplementary Table S7). Precision, recall and F1 score was always >0.89 for all downsampling datasets for the normal state, indicating that overall, the normal state was very robustly recovered regardless of sequencing depth. For the "gain" state, the precision and recall when 50% of the reads were retained were ~0.82 and ~0.65, respectively, meaning that there were fewer false positives than false negatives. False positives here are defined as new "gain" calls in the downsampled dataset in bins that were in a state "normal" or "loss" in the full dataset. Meanwhile, false negatives are bins that have lost their "gain" state in the downsampled dataset, to become either "normal" or "loss". Precision was lower for the "loss" state than for the "gain" state, while recall values remained similar (precision and recall were ~0.69 and ~0.65, respectively, at 50% coverage).

Overall, these results indicate that the genome-wide copy number gains and losses remained fairly stable upon downsampling even at

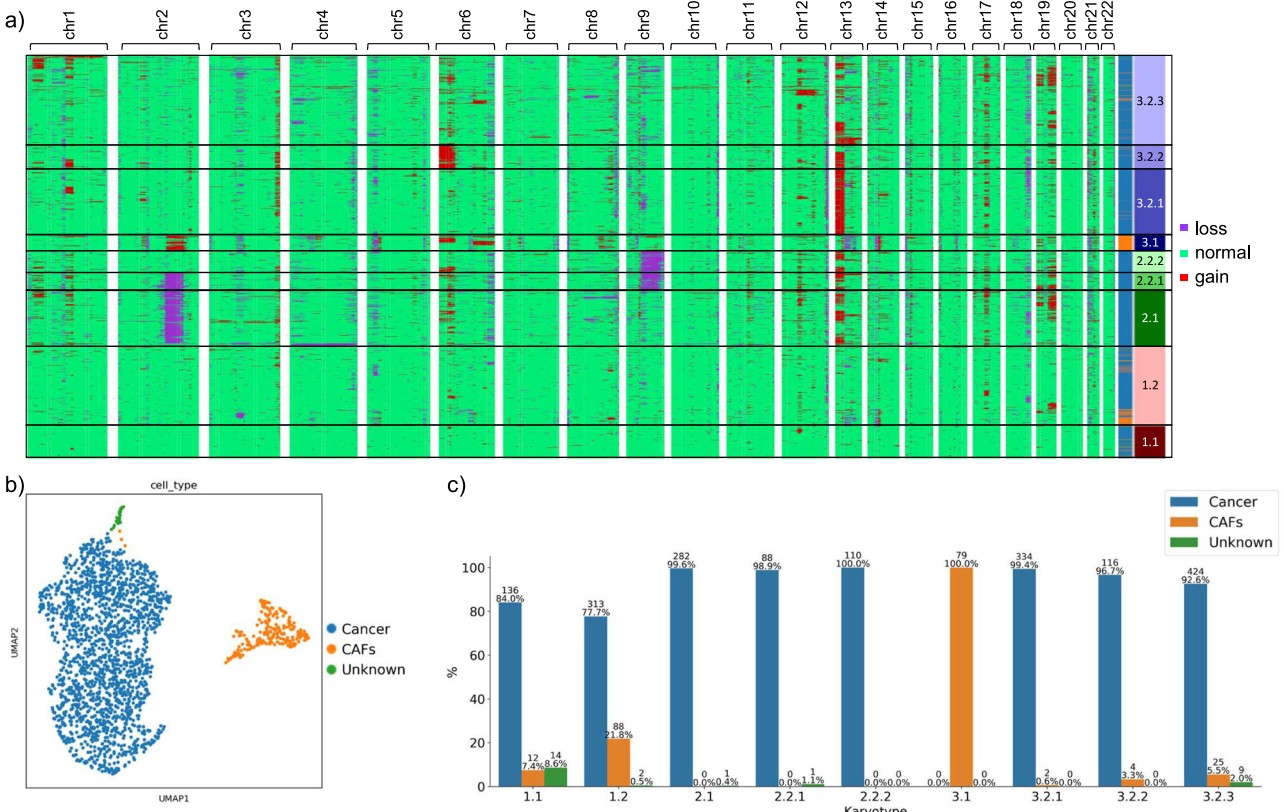

**Fig. 4 | Copy number alterations in a primary patient sample. a** CNAs for the SU006 primary sample obtained from scATAC-seq data. Every row is a cell, and every column is a chromosome. Karyotype clones are indicated in black boxes and numbered. Cell types are indicated per cell in the side bar. **b** UMAP embedding of the same cells showing cancer cells and cancer-associated fibroblasts. **c** Correspondence between karyotype clones from (**a**) and cell types from (**b**).

lower sequencing depths. The observed changes were mainly driven by "gain" states which were no longer identified in the low coverage datasets, as well as higher false positive and false negative "loss" calls stemming from the lower coverage in the subsampled data.

## epiAneufinder uncovers CNA tumor heterogeneity in primary patient samples

We further applied epiAneufinder to two patient samples of basal cell carcinoma that were profiled using scATAC-seq[18]. 2040 and 504 cells were profiled for two patients (named SU006 and SU008), respectively, with an average number of fragments of 58,055 and 63,060. The genome of both samples was binned into 100,000-bp bins and epiAneufinder was applied with standard parameters to identify breakpoints and assign copy number gains and losses per cell.

Several shared large gains and losses were identified in the genomes of both patients (Fig. 4a and Supplementary Figs. 10a and 11a). Patient SU006 presented one group of cells with a low number of copy number alterations (labeled cluster 1). Another group of cells was characterized by losses in chromosomes 2 and/or 9, and gains in chromosomes 13 and 19 (cluster 2). Another cell cluster showed the opposite, with losses in chromosome 13 and gains in 2, as well as in 6 (cluster 3.1). Finally, there was a cluster of cells characterized by gains of different lengths in chromosome 13 (cluster 3.2), and in chromosome 6 (cluster 3.2.2) or 19 (cluster 3.2.3). Patient SU008 also presented a nearly disomic cluster of cells (cluster 1), as well as a clone mainly characterized by different combinations of gains in chromosomes 1, 3, and 6 (cluster 2), and a cluster with combinations of a loss in chromosome 13 and gains in chromosomes 2, 6, and 19 (cluster 3).

Embedding the same cells based on their genome-wide scATAC-seq peak profile followed by Leiden clustering identified several clusters in both datasets. Differential gene activity levels for marker genes

allowed us to identify cancer cells (*KRT5, KRT15, CXCR4, TERT, TP63*), cancer-associated fibroblasts or CAFs (*COL1A2, LUM, FAP, VEGFC, ANGPT1, PDGFRB, IL6, CXCL8, CXCL12*) and endothelial cells (*CDH5, EGFL7*) (Fig. 4b and Supplementary Figs. 10b–e and 11b–e). We compared the Leiden clusters and the karyotype clusters (Fig. 4c and Supplementary Figs. 10f–i and 11f–i). For patients SU006 and SU008, CAFs corresponded mainly to the karyotype cluster 1.2–3.1 and 1.1–1.2–3.1–3.2, respectively, which shared similar CNA profiles between samples (Fig. 4a and Supplementary Fig. 11a). CAFs can harbor CNAs in their genome[36], and we identified karyotype clusters 3.1 (SU006) and 3.1–3.2 (SU008) as non-disomic. The cancer cells showed very different characteristics depending on the patient of origin: for patient SU006 the cancer was highly heterogeneous at the level of CNAs (Fig. 4c and Supplementary Fig. 10f–i), but the heterogeneity was much lower in patient SU008 (Supplementary Fig. 11f–i). The different CNA clusters present in each tumor could not have been identified based on the embedding results, also when the cells were embedded using the same windows used for CNA calling (Supplementary Figs. 10f–g and 11f–g), emphasizing the relevance of epiAneufinder for discovering new sources of variation in the data. In both patients, the mostly diploid karyotype clusters (cluster 1) contained a mixture of all cell types (Fig. 4c and Supplementary Fig. 11h), however all cancer cells showed upregulation of cancer markers, regardless of their karyotype (Supplementary Figs. 10e and 11e). These results highlight the power of epiAneufinder to identify novel sources of heterogeneity in the population of cells, based on their copy number profiles.

## Discussion

Copy number alterations characterize different human disorders, and have special relevance in cancer[4,37]. In particular, tumors often present CNA heterogeneity, with several clones contained in the same tumor

which may evolve differently during cancer progression and respond differently to treatment[1,38]. Measuring and quantifying the levels of CNA heterogeneity in cellular populations is therefore highly relevant[8]. Here, we present a computational method, epiAneufinder, which uses scATAC-seq data to faithfully recapitulate CNAs from single cells. This is achieved via segmenting the genome into equally sized bins and quantifying the number of reads mapping into every bin; to then test for significant differences in coverage depth along the genome. This is done by iteratively calculating the Anderson–Darling (AD) distance between the read distributions on each side of every bin. The highest AD distances mark the positions of the most likely copy number alteration breakpoints, based on which the genome is segmented. Every segment is then assigned to the state "normal", "gain" or "loss" based on its mean read coverage. The cells are finally clustered based on the similarity of their gain and loss profiles, to identify CNA clusters in the population.

Using several aneuploid cell lines and primary tumor samples, we showed how the genome-wide CNA profiles recovered from scATAC-seq data by epiAneufinder compared to the ones obtained from (sc)WGS, which can be considered a ground truth gold standard for the quantification of CNAs. EpiAneufinder was able to discover several clusters of cells in the populations, harboring different CNA profiles containing both gains and losses. EpiAneufinder was also applied to multiple scATAC-seq datasets from euploid samples, to show that it recognizes the absence of CNAs and correctly identifies euploidy.

Using datasets that contain multiple data modalities (scRNA-seq, scATAC-seq and (sc)WGS), we have compared epiAneufinder results to the ones from other algorithms that recover CNAs from scRNA-seq data (InferCNV, CaSpER and copyKat) or from scATAC-seq data (CopyscAT). In all comparisons, epiAneufinder outperformed the alternative methods, providing a genome-wide CNA profile most similar to the one obtained from (sc)WGS data. Finally, we also performed a simulation to study the robustness of epiAneufinder copy number gain and loss calls to different sequencing depths. As expected, for lower sequencing depths copy number gain identification becomes more challenging, while more copy number losses are wrongly called. However, the overall copy number profiles are recovered across the genome even at 40% of the initial coverage.

Having established the performance of epiAneufinder compared to (sc)WGS, we then moved to more physiologically relevant scenarios. Using two primary human samples of basal cell carcinoma that had been interrogated using scATAC-seq, we were able to identify distinct CNA profiles present in each tumor. This variation could not be revealed by embedding and clustering of the same cells based on their genome-wide scATAC-seq profiles. These results highlight the relevance of studying single-cell heterogeneity based on individual copy number profiles, which is otherwise hidden in classical single-cell ATAC-seq analyses.

In summary, epiAneufinder extracts single-cell copy number alterations from scATAC-seq data alone, or alternatively from single-cell multi-ome data, without the need to supplement the data with other data modalities. The method leverages read depth distribution along the genome to infer a separate karyotype for every individual cell, allowing to explore the CNA heterogeneity present in a sample at the single-cell level. EpiAneufinder unlocks a layer of genomic variation which is otherwise unexplored with traditional scATAC-seq data analysis, and its application offers the opportunity to explore relevant sources of heterogeneity which would otherwise remain hidden.

## Methods

### scATAC-seq, multi-ome, and (sc)WGS datasets pre-processing

The scATAC data for the gastric cell line SNU601 were aligned to the human genome hg38 using the 10x Genomics Cell RangerAtac 2.0.0 software with default parameters. The BCC samples as well as the PBMC and bone marrow samples (all part of the same study) were downloaded already aligned to the human genome version hg19. The COLO320HSR multi-ome dataset was obtained already aligned to the hg19 human genome version. The euploid brain samples and multi-ome samples[32] were aligned to the hg38 human genome version using the 10x Genomics Cell RangerAtac 2.0.0 software with default parameters for the scATAC and 10x Genomics Cell RangerArc 2.0 for the scATAC multi-ome set. The HCT116 cell line scATAC part of the multi-ome samples was aligned to the hg38 reference genome using the default parameters of STAR[39] after adapter trimming. Finally, the pediatric glioblastoma (pGBM) dataset was downloaded aligned to the hg38 human genome version.

The WGS dataset for the COLO320HSR cell line[30] was first trimmed in order to remove low-quality reads using the Trimmomatic software[40] (parameters PE, -phred33, ILLUMINACLIP:TruSeq3-PE.fa:2:30:10, LEADING:5, TRAILING:5, SLIDINGWINDOW:4:15, MIN-LEN:36). Then the reads were aligned to the hg38 human genome version, using the BWA-MEM aligner[41]. The aligned reads were further processed in order to remove duplicates using the MarkDuplicates software from the Picard toolkit (https://broadinstitute.github.io/picard/). The WGS data of the pediatric glioblastoma samples (pGBM) were downloaded already aligned to the hg38 human genome version.

The scWGS of the SNU601 and the HCT116 cell lines were downloaded already aligned to the hg38 genome.

### CNA calling of (sc)WGS data and comparison with scATAC

The scWGS of the SNU601 cell line[23] and HCT116[42] were analyzed using aneufinder[24] with default parameters and window size 100 kb. For the SNU601 cell line, aneufinder identified a cluster of cells with very high ploidy, cluster 2, consisting of 134 cells (Supplementary Fig. 3), which we excluded from the scWGS data for the comparisons between scATAC and scWGS. Since the cell lines were measured in different laboratories, clonal variation among them can be expected[25].

For the processed WGS datasets, aneufinder was used to calculate GC-normalized reads per bin. These reads were then normalized using the median value for the dataset and the resulting signal was used as the ground truth. For the comparison with the epiAneufinder pseudo-bulk results, the WGS signal was smoothed using a gaussian filter and the scATAC and WGS signals were standardized (mean=1 and std=0) in order for the two signal to be placed in the same scale.

For the comparison between the two different data modalities in each of the two datasets, first we retained the same windows in both modalities. Afterward, for the single-cell data, a pseudo-bulk CNA profile was generated by counting the number of gain/loss/normal cells identified per window, and multiplying each gain by 3, each loss by 1 and each normal state by 2:

$$\overline{CNV}_i = \frac{1}{N_{cells}}\left(\sum_{N_n} 2 + \sum_{N_l} 1 + \sum_{N_g} 3\right), \qquad (3)$$

where $N_{cells}$ is the total number of cells, $N_n$ is the number of "normal" cells, $N_l$ is the number of "loss" cells and $N_g$ is the number of "gain" cells, at bin $i$. When comparing scATAC-seq to WGS, we standardized (mean=1 and std=0) the values in both modalities to place them on the same scale. Pearson correlation was calculated between the ATAC and the DNA profiles, as well as the Mean Square Error (MSE), with:

$$MSE = 1/N_{bins}^2 * \sum_i \left(\overline{CNV}_i^{DNA} - \overline{CNV}_i^{ATAC}\right)^2$$

calculated between the scATAC and the scWGS profiles, both genome-wide and per chromosome. Precision, recall and F1 scores were also calculated for the comparison between the ATAC and the DNA profiles. For the comparisons of single-cell ATAC versus single-cell WGS profiles (SNU601 and HCT116 cell lines) two thresholds $c$ were used to convert

the data into a classification problem: in each dataset separately, bins with mean copy number above $c_{high}$ were called gain, bins with mean copy number below $c_{low}$ were called loss, and the rest were called normal. For the comparison between single-cell ATAC results and bulk WGS, first the WGS dataset was converted into a classification (every bin was called normal, gain or loss given two fixed thresholds) and then two thresholds were used to generate the ATAC classification as described above. For every value of $c_{high}$ and $c_{low}$, the two classifications were compared to calculate the precision, recall and F1 scores for every state.

### Embedding of primary samples based on scATAC-seq profile

Peaks were called using MACS2[43] on the aggregated dataset. The epiScanpy toolkit[44] was used for subsequent analysis. First, we filtered observations to match the barcodes from the epiAneufinder output, built a peak matrix, as well as a gene activity matrix. Then the peak matrix was binarized, variability scores (as defined by epiScanpy) were calculated per feature and the features were selected based on a variability score threshold of 0.53. Prior to further analysis the peak matrix was library size normalized and logarithmized. The gene activity matrix was library normalized. Principal component analysis was performed and the most informative PCs were selected using the elbow method (SU006: 6, SU008: 5). Using these top PCs, we computed a neighborhood graph and an embedding using UMAP. The Leiden community detection algorithm was used for clustering and we identified the top differentially open peaks between clusters which were used for enrichment analysis with GREAT[45] (Supplementary Fig. 7). Moreover, several marker genes for cancer cells, fibroblasts and endothelial cells were quantified per cluster[18,46] (Supplementary Fig. 7). After evaluating marker gene activity and GO terms, clusters were annotated. We finally computed the composition of the cell type clusters with respect to the karyotypes, and vice versa.

### Comparison to other single-cell CNA calling methods

We compared epiAneufinder to three methods developed for CNA calling from scRNA-seq, inferCNV[16], CaSpER[12], and CopyKat[14], and one method developed for CNA calling from scATAC-seq, Copy-scAT[28]. For all methods, the analysis was performed by running the respective method on default parameters.

The scRNA-seq-based methods were run on cell lines SNU601, HCT116, and COLO320 replicate 1. All methods required a matched reference dataset to call CNAs from scRNA-seq data. We used for the SNU601 cell line control samples from a gastric cancer study[33], for the HCT116 cell line non-tumor tissue samples from colorectal cancer patients[34] and for the COLO320 cell line healthy adult samples from the gut cell atlas[35]. For CopyKat and InferCNV, the count matrices were taken directly from GEO for the analysis. For CaSpER, the additionally necessary bam files were generated by mapping the fastq files with 10x Genomics Cell Ranger 7.0.0 in case of the 10x datasets (SNU601 and COLO320) or with STAR[39] 2.7.10b in case of the Smart-seq2 dataset (HCT116). We evaluated the correlation between the genome-wide results of each RNA method compared to the (sc)WGS results using pseudo-bulk aggregates. For this, the gene-based CNA results of the RNA methods were mapped to the genomic bins of the (sc)WGS and epiAneufinder results, averaging the CNA scores when multiple genes fall into the same bin. Only bins where results were available for all methods were included in the comparison, i.e., a far smaller part of the genome is evaluated compared to using only scATAC-seq data.

The scATAC-seq-based method Copy-scAT was run on cell lines SNU601 and COLO320 replicate 1. By default, Copy-scAT reports CNAs on chromosome-arm level, which were transferred onto the epiAneufinder bins. This allowed us to calculate pseudo-bulk-level correlations across bins between Copy-scAT, epiAneufinder, and (sc)WGS.

### Downsampling of scATAC dataset and analysis

The SNU601 cell line dataset was downsampled with the 10x Genomics Cell RangerAtac 2.0.0 software[18] using the count command (subsample rate from 0.1 to 0.9). The results of the downsampling process can be viewed in Supplementary Table 3. EpiAneufinder was run for each percentage of the original dataset, with the same parameters as for the full set. In the filtering step of the algorithm, both cells and bins that did not pass the epiAneufinder quality controls were removed (Supplementary Table 4). To compare the downsampled datasets with the full dataset, only common cells and bins were considered. For these, we quantified the number of bins that did not change their CNA status and the number of bins that changed their status from gain/loss/normal to a different one (Fig. 4).

To further explore the robustness of the algorithm, we calculated "precision", "recall" and "F1" scores using as a ground truth the CNA calls from the fully covered dataset (Supplementary Table 5). For the gain state, we defined as true positive (tp) the bins with gain state in both the downsampled and the full set, as false positive (fp) the bins with gain in the downsampled but not in the full set and false negative (fn) the bins with gain in the full set but not in the downsampled one. Similarly, we defined true positives, true negatives, and false negatives for the loss and disomic states. Then the precision, recall, and F1 scores were calculated as:

$$Precision = \frac{tp}{tp + fp} \tag{4}$$

$$Recall = \frac{tp}{tp + fn} \tag{5}$$

$$F1 = 2 * \frac{Precision * Recall}{Precision + Recall} \tag{6}$$

### Statistics and reproducibility

No statistical methods were used to determine sample size. No data were excluded from the analysis.

### Reporting summary

Further information on research design is available in the Nature Portfolio Reporting Summary linked to this article.

## Data availability

The following publically available datasets analyzed in this study were downloaded: The scATAC-seq dataset for the SNU601 cell line[19] was downloaded from the Short Read Archive (SRA) accession "PRJNA674903" and the scWGS data for the SNU601 cell line[23] was downloaded from SRA accession "PRJNA498809"; the scRNA samples for the SNU601 cell line[23] were downloaded from GEO under accession number "GSE142750" and the control samples[33] from "GSE150290" (only the normal stomach) for the scRNA CNV calling. The two pre-treatment basal cell carcinoma samples, the PBMC and bone marrow euploid samples[18] were obtained from SRS accession "GSE129785" (accession number GSM3722057 for SU006, GSM3722064 for SU008, GSM3722015 for PBMC and GSM3722071 for bone marrow). The multi-ome and scATAC brain samples[32] were downloaded from the Gene Expression Omnibus (GEO) database, series number "GSE162170" (accession numbers for the multi-ome GSM5584685, GSM5584686, GSM5584687 and for the scATAC GSM4944156, GSM4944157, GSM4944158 and GSM4944159). The pediatric glioblastoma[28] scATAC data were downloaded from GEO, under accession numbers "GSE163655" and "GSE163656", while the matching WGS data[31] were downloaded from the European Genome-Phenome Archive (ENA), under accession number "EGAD00001005212". The multi-ome dataset

of the COLO320HSR cell line[27] were downloaded from the SRA accession "PRJNA672109", while the WGS of the same cell line[30] were downloaded from the SRA accession "PRJNA506071" (sample SRS4831935) and the control scRNA samples[35] from the https://www.gutcellatlas.org/. The scWGS dataset of the HCT116[42] was downloaded from the European Nucleotide Archive (ENA), under accession number "PRJEB27084" and the multi-ome dataset[26] was downloaded from the SRA with accession number "SRP167062 [https://trace.ncbi.nlm.nih.gov/Traces/index.html?view=study&acc=SRP136421]". For the scRNA CNV identification, as a control sample was used a set of non-cancer samples[34] downloaded from GEO under the accession "GSE146771". All other data supporting the findings of this study are available within the article and its supplementary files. Any additional requests for information can be directed to, and will be fulfilled by, the lead contact. Source data are provided with this paper.

## Code availability

epiAneufinder is available through Github (https://github.com/colomemaria/epiAneufinder, https://doi.org/10.5281/zenodo.8032096).

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

## Acknowledgements
We would like to thank Dr. Antonio Scialdone for discussions about calling CNAs in scRNA-seq data, and Anna Danese for help in embedding of scATAC-seq data. We thank Dr. Andres Castellanos for the discussions regarding the BCC tumor composition. We thank Dr. Kainat Khowaja and Ronan Le Gleut from the Core Facility Statistical Consulting at Helmholtz Munich for statistical advice. We thank the BMC Bioinformatics Core Facility for providing access to their HPC cluster. This work was supported by the Impuls-und Vernetzungsfonds of the Helmholtz-Gemeinschaft (grant VH-NG-1219) for M.C.T. and A.R. A.S. acknowledges support from HelmholtzAI. P.H. is supported by the Helmholtz Association under the joint research school "Munich School for Data Science — MUDS".

## Author contributions
M.C.T. designed the study. M.C.T. and A.R. conceived the algorithm. A.R. and A.S. implemented the algorithm. A.R., A.S., P.H., K.T.S., and M.L.R. analyzed the data. M.S. provided code. All authors contributed to manuscript writing.

## Funding

## Competing interests
The authors declare no competing interests.
