## [Peer Review File · Nature Communications]

REVIEWER COMMENTS

Reviewer #1 (Remarks to the Author):

In this manuscript, Ramakrishnan et al. present a new algorithm, epiAneufinder, to generate genome-wide copy number profiles of single cells based on the read counts from single-cell ATAC-seq data.

The new algorithm is a useful addition to the expanding arsenal of computational tools to assess copy number variation and aneuploidy from single cell-derived data. The ability to accurately infer CNVs/CNAs from scATACseq could add an extra layer of genetic information to scATACseq experiments. The logic behind the algorithm seems sound. Two additional advantages are that there is no need for a reference euploid sample, and that the algorithm is provided as a ready-to-use R package (which seems to be user-friendly). While I support publication in Nat Commun, I think that the study should be expanded prior to its acceptance, in order to strengthen the confidence in the new algorithm and better compare its performance to that of competing approaches.

As I am not an scATACseq expert, my comments/suggestions below focus on the biological relevance and the performance of this new tool:

(1) A major challenge that the algorithm attempts to cope with is the coverage sparsity that is inherent to single cell sequencing. Certain genomic locations, such as telomeric and centromeric regions are removed due to the difficulty to map them. Is it possible to use the new telomere-to-telomere (T2T) genome reference in order to 'rescue' such regions and thus increase the number of available reads and their genomic coverage?

(2) The agreement between scWGS data and scATACseq data, as shown in Fig. 2 in the SNU601 cell line, appears to be reasonable. Are there additional cell lines that can be compared in such a way? It is impossible to tell whether this one case is representative or not. In addition to the need to add a couple of additional examples, it also seems important to run the algorithm on scATACseq data from cells known to be euploid and chromosomally stable (e.g., any scATACseq dataset from normal primary human cells) – to evaluate the false discovery rate (as a function of genomic resolution) of the algorithm. The authors attempt to do this to some degree using simulations (Fig. 4), but real-world data from stable euploid cells would be preferable.

(3) The relatively low degree of agreement between the WES and the scATACseq-inferred CNVs (Fig. 3) is somewhat concerning. Pearson's correlations of ~0.5-0.6 are quite low. The authors state that "this level of agreement is notable given the sharp differences between the two data types and the different areas of the genome that they cover." However, this level of agreement isn't really acceptable, so it seems important to find out which of the two methods better represents the actual CNV landscape of the tumor.

(4) Related to the previous comment – WES is being compared to epiAneufinder in tumor samples, whereas scWGS is being compared to epiAneufinder in cancer cell lines. Could the level of agreement be affected by the nature of these samples? Ideally, both WES and scWGS would be compared to the new algorithm both in cell lines and in primary tumors (separate comparisons, but same type of samples).

(5) Lastly, the comparison to DNA-based sequencing approaches (scWGS and WES) are indeed the most important comparisons. Nonetheless, it would be highly valuable if the authors could compare their algorithm to those that infer CNVs from scRNAseq data (inferCNV, CaSpER, HoneyBADGER). While inferring CNVs from scRNAseq data can be challenging, as noted by the authors, it has pro's and con's compared to the scATACseq data, and a head-to-head comparison would be informative.

Reviewer #2 (Remarks to the Author):

Summary

epiAneufinder is a tool to detect CNVs from scATAC-seq data. It uses a binary segmentation algorithm to locate CNV breakpoints for each cell by calculating the AD distances between read distributions of neighbouring bins/segments and assigning a putative breakpoint to the genomic position with max AD. Breakpoints are pruned if they show an AD distance lower than the genome-wide mean and copy number is calculated for each bin as "loss", "gain" or "normal" depending on deviation from this mean. The authors apply the tool to characterize an adenocarcinoma cell line (SNU601), and two basal cell carcinoma patient samples (SU008 and SU006) in order to characterize the karyotype and clone diversity in these published datasets. The paper is well written and clear, however, I believe substantial benchmarking is still required (critical to actually assess performance), and there is a lack of interpretation and biological meaning assigned to the genetic subclones described in the study.

Major comments

By comparing against genome-wide mean coverage, the tool assumes a diploid/normal baseline karyotype for the sample and does not consider for instance whole genome doublings, which are indeed common to cancers. This should be discussed.

To define CNV breakpoints a binary segmentation method iteratively calculates AD for each segment until a “stopping criteria” is reached. This is by default set to seven (line 117). The value seems arbitrary. Why did the authors select this value as default? Is the same value applied to every chromosome? If so, how does it account for chromosomes of differing lengths, and do shorter chromosomes suffer too many breakpoints? How does changing the value alter CNV predictions?

Additionally, after initial segmentation, the breakpoints are “pruned” if the AD distance is lower than the genome-wide mean (line 119). This seems to imply the authors assume a mostly ‘normal’ or diploid background for each input cell, which may indeed be erroneous and lead to false negatives. e.g. whole genome doubling events will escape detection, which are common in cancers. This should be clearly discussed

It seems this breakpoint approach is prone to error, which is likely due to the genome being ‘artificially’ binned prior calculating AD – this is especially clear in Fig S1 where several “loss” bins have a read density similar to the “normal” distribution (hidden behind the normal/gain areas of s1a and visible in s1c). Can a breakpoint refinement approach (e.g. sliding window, and/or pseudobulk clustering) be implemented to better define a more real CNV location?

What is the size distribution of CNVs identified? How balanced are the “gains” to “losses”? what is the limit of detection?

The heatmap and clone composition shown in e.g. Fig 2/S2 contains considerable mixing of karyotypes (e.g. mentioned on line 156). How do the authors interpret the subclone structure? It is reasonable to think the same region would delete multiple times in each subclone independently? Or are these false calls? The authors should discuss this.

There is little meaning or biological analysis of the clones defined. Why are some cells grouped into one clone when they show very diverse karyotypes? Are there specific features driving this, or any genes/pathways implicated? Overall, an interpretation of the clone structure and biological relevance is missing for all samples.

Looking at the pseudo-bulk CNV values of Fig 2c, it is hard to believe they are “highly correlated” with a value of 0.86. There are very clear differences between the karyotypes identified in each sample, and

this feels ignored. As briefly mentioned in text (lines 180-6), several areas are not called consistently between the modalities (including e.g. losses on chr9q often 'missed' by epiAneufinder). Can you show the correlation value for every bin? Why do some chromosomes or regions appear to 'suffer' more than others?

How does gene activity change the CNV calling? Can the authors test how CNV gains/losses correlate with number of highly/lowly expressed genes per segment?

This validation approach fails to consider the accuracy of CNV calls in each individual cell and comparing to the scWGS data as the "ground truth" is confounded by the clone diversity. I do appreciate having a perfect 'truth set' is nearly impossible here, but I think more benchmarking is needed and comparison of 'true' CNVs called per cell is missing. Comparing outputs against other CNV tools can help with this.

Can the authors identify the most-likely shared clone between the ATAC and scWGA datasets of Fig2 and compare the CNV recall rates between these (pseudo-bulk'ed) cells?

Is there a scRNA-seq + scATAC-seq dataset that can be included? This might be useful for performing the benchmarking. Despite CNV-calling from scRNA-seq being rather imperfect, this could give a better head-to-head comparison of precision/recall per cell.

The poor correlation seen between scATAC and WES (line 213) is a little alarming. Can the authors adjust parameters (e.g. binning) to improve this? Would it be an option to compare the consistency between the most-frequent CNV status/bin of the sc data to the bulk WES data?

In Fig 3 the "fibroblast" classified cells (clone 1.1) show more genetic abnormalities than some "tumor" cells (clones 1.2, 1.3). Why is that?

Since clone composition appears extremely diverse and the clones are not fully characterized, I have trouble with some of the claims, for example on lines 223-227. The authors say that SU006 contains 8 karyotype clones, but the clone borders do not appear to define karyotypes (see above comment). And indeed clone 1.2 appears karyotypically normal, but is still being called "tumor". Moreover, clone 3.1 in SU008 contains a huge gain on chr6 that is shared with tumor cells but is assigned as fibroblast – how do authors interpret these inconsistencies and what is the biological reasoning behind this?

As I understand CNV calls are very much dependent on depth of coverage. This is true for all calling algorithms, even on bulk data. Thus I have reservations about the down-sampling experiment results. Is there surprisingly high performance at the 20% subset- is this because they are only comparing bins that are shared with 100% set? How many bins were lost during the down-sampling? Why is the performance not reducing at expected rates? How many clones are captured with each subsampling, how many are lost?

The authors claim that after down-sampling the overall copy number profiles are recovered across the genome (line 300) but these data are not shown, and hard to believe.

Minor comments

CNVs are very broadly defined as => 2 are “gains”, can better resolution of copy number values be achieved?

precision and recall should be plotted as curves

Fig1A – is this real single-cell ATAC-seq data? For which cell? How representative is this to expected coverage for scATAC-seq? I ask, because in the manuscript it says the median coverage is ~75k fragments/cell, thus with the genome binned into 100kb segments I would expect ~2-3 reads/window, but instead ~10 is shown for “normal”.

Fig1B – the CDF panels on the far right are not explained and unclear what they are showing

Fig S1 appears to be a replicate of Fig 1 – would be nice to see also other representative cell(s) with differing karyotypes

Fig S2 (etc) - The clones should also be indicated, to provide reference with the dendogram

Fig s3b - I found this file unmanageable, and am not sure what it is showing. Is there option to reduce size and improve readability.

Fig 2c – axis is not labelled.

Fig s8 – the clone colors are not consistent between a and b – it would also be easier to read if the colors were darker and clone-themed (e.g. 3.1-3.4: shades of green vs 1.1-1.3: shades of blue)

The sentence on lines 230-232 seems to overstate claims. There tumor cells were not identified by their copy number profiles, as I believe that is why a classifier was used. Also there appears to be little relationship between the karyotypes of “tumor cell” clones, and very little concordance between the UMAP clusters and clones, so I do not believe this sentence is supported by data.

Reviewer #3 (Remarks to the Author):

The authors describe epiAneufinder, a tool that enables the detection and profiling of copy number alterations from single-cell ATAC-seq datasets. This tool, similar to methods that leverage scRNA data for CNA calling, seeks to gain additional information out of tumor samples other than the primary modality. The tool fills a very important need – often samples have epithelial cell types that are difficult to distinguish between tumor-adjacent epithelial cells or cancer cells, and a measure of aneuploidy is a powerful way to make this determination. Therefore the impact of the work is high and I expect there will be a great deal of use by the method. While the results are somewhat noisy – they are clean enough to aid in meaningful biological interpretation and the tool marks a nice step forward an analytical methods for ATAC data.

Note: CNV is typically used to encompass germline variation within populations as opposed to CNA (copy number alteration) which is more general and applicable to cancer, where this tool is most appropriate. I don't think it matters really, since papers often use them interchangeably, so it is just a suggestion.

A notable strength is that the tool is very easy to implement. We did not have any challenge running it according to the documentation on our own datasets on the first try. This is in contrary to many other tools that get released.

The results appear to be fairly noisy – we also observed this on our datasets; however, we were able to extract general trends – ie cells from a certain cluster are more frequently showing a certain aneuploid profile, thus allowing us to identify that cluster as cancer vs tumor-adjacent epithelial. A useful function would be the addition of a metric for “estimated probability of aneuploidy”. This would be very useful, as one could cross-reference with identified clusters based on ATAC signal and perform this discrimination and it would not rely on having really high coverage ATAC data to get granular copy

number calls. (not a requirement for publication, just a suggestion of something that would be super useful in many cases) Overall, the noisiness is much lower than I would expect from ATAC-based CNV calling, so the tool appears to work quite well – this suggestion is more to accommodate noisier datasets as well.

On the primary patient samples, all cells are ‘tumor’ since the sample is from a tumor – the cells labeled as such are cancer cells / neoplastic cells within the tumor.

Is it possible to normalize against a sample of known euploid cells? i.e. a known population of cells with a similar cell type to a suspected neoplastic cell cluster – and then get cleaner results?

Are the authors certain that the cancer cell population – subclone 1 – are actually cancer cells? They could just be non-neoplastic epithelial based on the ATAC analysis. Also there are no immune cells identified which is odd to me...

In figures S8 and S9 I would expect to see some more consistency in the karyotype vs ATAC clusters. In the ‘tumor’ cluster the karyotypes appear randomly distributed... I would expect that distinct subclones would have some shifts in their epigenetic profiles; but this may not necessarily be the case.

Point-by-point response to the reviewers' comments

“epiAneufinder: identifying copy number alterations from single-cell ATAC-seq data”

Akshaya Ramakrishnan, Aikaterini Symeonidi, Patrick Hanel, Katharina T. Schmid, Maria L. Richter, Michael Schubert, Maria Colomé-Tatché

We would like to thank the editor and the reviewers for their comments on our manuscript. We have fully addressed all the concerns and we provide here a point by point answer (in blue) to the reviewer's comments (in black). The line numbers correspond to the new version of the manuscript with highlighted changes.

Among other improvements, we have added several new samples to our analysis, including cell lines and primary tumors. We are very pleased to have added additional primary samples in our analyses, instead of focusing on cell lines only. Unfortunately, the legal agreements for this data transfer delayed our manuscript resubmission by several months, for which we would like to apologize. Nevertheless, we think the addition of these samples makes our manuscript more valuable and interesting to a broad audience.

Reviewer #1 (Remarks to the Author):

In this manuscript, Ramakrishnan et al. present a new algorithm, epiAneufinder, to generate genome-wide copy number profiles of single cells based on the read counts from single-cell ATAC-seq data.

The new algorithm is a useful addition to the expanding arsenal of computational tools to assess copy number variation and aneuploidy from single cell-derived data. The ability to accurately infer CNVs/CNAs from scATACseq could add an extra layer of genetic information to scATACseq experiments. The logic behind the algorithm seems sound. Two additional advantages are that there is no need for a reference euploid sample, and that the algorithm is provided as a ready-to-use R package (which seems to be user-friendly). While I support publication in Nat Commun, I think that the study should be expanded prior to its acceptance, in order to strengthen the confidence in the new algorithm and better compare its performance to that of competing approaches.

As I am not an scATACseq expert, my comments/suggestions below focus on the biological relevance and the performance of this new tool:

(1) A major challenge that the algorithm attempts to cope with is the coverage sparsity that is inherent to single cell sequencing. Certain genomic locations, such as telomeric and centromeric regions are removed due to the difficulty to map them. Is it possible to use the new telomere-to-telomere (T2T) genome reference in order to 'rescue' such regions and thus increase the number of available reads and their genomic coverage?

We thank the reviewer for this suggestion. We would like to clarify that missing regions in the genome assembly are not an issue for CNV calling. Rather, the confidence of the mapping could mislead the CNV calling procedure, i.e., regions that cannot be mapped with high enough confidence are not suitable to quantify copy number variations. Including repetitive regions of the genome, by using for example a T2T reference, would therefore unfortunately not help with calling CNVs in these regions.

Overall, by removing black listed regions, we only discard a small percent of the genome, ~13% to be exact.

(2) The agreement between scWGS data and scATACseq data, as shown in Fig. 2 in the SNU601 cell line, appears to be reasonable. Are there additional cell lines that can be compared in such a way? It is impossible to tell whether this one case is representative or not.

We thank the reviewer for this comment which adds strength to the results presented in our manuscript. We have obtained further datasets where the scATAC-seq profiles can be compared to the ones from whole genome sequencing (WGS) or single-cell whole genome sequencing (scWGS). We have included cell lines, but also primary samples where we have an orthogonal confirmation of the copy number variations:

-HCT116 cell line ([1] and [2] for scATAC-seq and scWGS respectively)

-Colo320 cell line ([3] and [4] for scATAC-seq and WGS respectively)

-Primary and relapse patient pediatric glioblastoma dataset ([5])

To compare the results between scATAC-seq and whole genome sequencing we now compute the precision, recall and F1 of the calls using the (sc)WGS as a ground truth. We do that to better compare the two results, instead of using correlations (see please answer to reviewer 2, point 7 below).

In all the analyzed datasets, comparing the epiAneufinder CNVs to the (sc)WGS led to very high values of the precision, recall and F1. Note that when a state is not present in the dataset, the F1 value is either NA or very low (as we are comparing noise vs noise).

This can be visualized with the karyotypes for all datasets, as well as the genome-wide pseudo bulk profiles:

HCT116 cell line:

	precision	recall	F1
loss	0.623946037099494	0.501355013550135	0.555972952667167
normal	0.983336096833029	0.988169624260602	0.985746935383337
gain	0.95875	0.975206611570248	0.966908288685786

Reviewer response Figure 1: HCT116 cell line: scATAC-seq karyotype (top), scWGS karyotype (middle), and pseudo bulk comparison (bottom). The table shows the best precision, recall and F1. Note that the pseudo bulk for the sc genome sequencing (GS) is calculated from only 22 cells, hence the jagged single cell genome sequencing profile, especially seen in the small sharp losses which are mostly due to noise in the coverage.

Colo320 cell line:

	precision	recall	F1
loss	NA	NA	NA
normal	0.933056278790306	0.922205176424011	0.927598994499614
gain	0.649810648251281	0.685707569346497	0.667276678485646

Reviewer response Figure 2: Colo320 cell line: scATAC-seq karyotype (top) and pseudo bulk comparison to WGS data (bottom), for replicate 1. The table shows the best precision, recall and F1 (no losses identified). The karyotype for all replicates can be found in SI Figure 5.

Pediatric glioblastoma:

	precision	recall	F1
loss	0.774386594853381	0.889347079037801	0.827895073576456
normal	0.927005851433105	0.925033400959968	0.926018575851393
gain	0.723703344643723	0.694580134915097	0.708842729970326

Reviewer response Figure 3: Primary patient glioblastoma dataset: scATAC-seq karyotype (top) and pseudo bulk comparison (bottom) to WGS data, for replicate 2937. The table shows the best precision, recall and F1. All replicates can be found in SI Figure 6.

These results have been added to the section “epiAneufinder CNAs are concordant with the ones obtained from (sc)WGS” (line 147) and to the SI Figures 4, 5 and 6.

As a side note: the high throughput 10x scWGS kit (called single-cell CNV method) is not sold any more by the 10x company. Because of this, there are not a lot of

high-throughput new datasets that quantify CNVs using single-cell whole genome sequencing, as they require an in-house protocol.

(3) In addition to the need to add a couple of additional examples, it also seems important to run the algorithm on scATACseq data from cells known to be euploid and chromosomally stable (e.g., any scATACseq dataset from normal primary human cells) – to evaluate the false discovery rate (as a function of genomic resolution) of the algorithm. The authors attempt to do this to some degree using simulations (Fig. 4), but real-world data from stable euploid cells would be preferable.

We thank the reviewer for raising this very important point. To address it, we have now run epiAneufinder in 9 euploid samples from different sources:

-Brain single-cell ATAC-seq (4 replicates) and single-cell multi-ome (3 replicates) datasets [6]

-Bone marrow (1 replicate) and PBMC (1 replicate) scATAC-seq data [7]

In these euploid datasets analyzed, epiAneufinder does not find relevant copy numbers. There are a few non-shared small CNVs found in some cells, which can be most likely attributed to noise, as can be seen in the karyotypes for the normal cells shown below:

a)

b)

c)

d)

e)

f)

g)

h)

i)

Reviewer response Figure 4: a-d) Karyotypes for the scATAC-seq brain samples (top), and pseudo bulk results compared to a baseline of “normal”. e-g) Karyotypes for the ATAC-seq modality from the multi-ome data for brain samples (top), and pseudo bulk results compared to a baseline of “normal”. h) Karyotypes for the scATAC-seq PBMC sample (top), and pseudo bulk results compared to a baseline of “normal”. i) Karyotypes for the scATAC-seq bone marrow sample (top), and pseudo bulk results compared to a baseline of “normal”.

Assuming a ground truth of “normal” and comparing our pseudo bulk scATAC-seq results to it (aggregate of the CNV value per cell in every bin, divided by the number of cells) yields a recall of

Sample	Recall
Satpathy_BoneMarrow	0,97459945
Satpathy_PBMC_rep1	0,98485799
Greenleaf_brain_rep1	0,97002464
Greenleaf_brain_rep2	0,97008819
Greenleaf_brain_rep3	0,9710737
Greenleaf_brain_rep4	0,97130869
multiome_brain_rep1	0,98151908
multiome_brain_rep2	0,98316854
multiome_brain_rep3	0,98261042

Please note that the precision cannot be calculated because we only have true positives and false positives in that dataset. In addition, we have also calculated the variation and the mean square error of these results (mean square error considering a ground truth of “normal”), to show how much deviation there is between our results and the expected “normal” result genome wide. These values are very low, showing that we indeed have not discovered any relevant copy numbers:

Sample	Variation	MSE
bone marrow	0,01251366	0,00067758

PBMC	0,00921701	0,00038248
brain rep1	0,01485453	0,00093185
brain rep2	0,01436002	0,00085472
brain rep3	0,01440061	0,00085159
brain rep4	0,01458655	0,00088512
multiome brain rep1	0,01408168	0,00079308
multiome brain rep2	0,013363	0,00071464
multiome brain rep3	0,01374836	0,00075774

With these results, we show that epiAneufinder recognizes absence of CNVs and correctly identifies euploid samples.

We have added the results on the euploid samples in the manuscript in section “epiAneufinder CNAs are concordant with the ones obtained from (sc)WGS” (line 147) and to the SI Figure 7 and SI Table 4.

(4) The relatively low degree of agreement between the WES and the scATACseq-inferred CNVs (Fig. 3) is somewhat concerning. Pearson’s correlations of ~0.5-0.6 are quite low. The authors state that “this level of agreement is notable given the sharp differences between the two data types and the different areas of the genome that they cover.” However, this level of agreement isn’t really acceptable, so it seems important to find out which of the two methods better represents the actual CNV landscape of the tumor.

We agree with the reviewer that the level of agreement with the WES data is, despite being non-negligible, low.

We have investigated this issue further. For the dataset that we used, we cannot know what is the real CNV landscape of the tumor, as we have no alternative measurement for it (FISH, karyotyping or other). We have however analyzed the WES data with two widely used CNV callers that have optimized parameters for CNV calling on WES data, namely CNVkit [8] and Control-Freec [9].

We found very different CNV results depending on the method used: for tumor 006, Control-freec identified 68 loss regions and 3 gain regions. The loss regions had a total number of 1,968,809,062 bases (almost 2Gbases) and the gain regions 94,165 bases. For the same tumor, CNVkit identified only 56 loss regions for a total of 2.5Gbases, and no gains.

For tumor 008, Control-freec identified 4 loss regions and 51 gain regions. The loss regions had a total number of 5,157,912 bases (5Megabases) and the gain regions

298,637,231 bases (298Mbases). For the same tumor, CNVkit identified 55 loss regions for a total of 2.5Gbases.

For both tumors, there was some overlap between the losses that were identified by CNVkit and the ones from Control-freec, but overall Control-freec identified more “lost” regions. In addition, CNVkit was unable to identify any regions of gain, while Control-freec identified a few gain regions.

Despite our initial surprise, these results corroborate the ones from a recent benchmarking paper from Gabrielaite et al. [9]. In that article, the authors evaluate the performance of CNV calling methods from WES data and conclude that “*In summary, by reviewing 50 tools for CNV calling, of which 11 were included for a benchmark (CLC Genomics Workbench (WGS and WES), cn.MOPS (WGS and WES), CNVkit (WES), CNVnator (WGS), CODEX2 (WGS), Control-FREEC (WGS), DELLY (WGS), ExomeDepth (WES), GATK gCNV (WES and WGS), Lumpy (WGS), and Manta (WES and WGS)), we conclude that CNV identification from NGS data remains challenging. For the best reliability of CNV calling from NGS data, we observed that even if the tools were developed for WES data or allowed it as input, they did not perform well. We suggest WGS as the only NGS-based option for broad calling of CNVs. Furthermore, low precision in all tools leads us to recommend a hypothesis-based approach for finding causative CNVs by NGS in the clinic, and further validation of these candidates by manual inspection, MLPA or array-based approaches.*”. Our results are therefore in agreement with their conclusions.

After our analysis and comparison with the benchmarking paper cited above, we unfortunately concluded that WES is not the best data modality to use as ground truth for CNV calling. We would like to apologize for not realizing this beforehand, and thank the reviewer again for raising the concerns that led to this realization.

We have therefore removed the comparison to the WES results, which cannot be used as ground truth. Nevertheless, we have decided to keep the scATAC-seq CNV results from the BCC dataset in the paper, without orthogonal ground truth. Based on our comparisons between epiAneufinder results and the ones from (sc)WGS data, we find it safe to assume that epiAneufinder works robustly for calling CNVs from scATAC-seq data. Considering the extra datasets that we discussed in points (2) and (3) above, we have now in total seven successful comparisons to (sc)WGS data (with multiple technical and biological replicates), including both euploid (brain, PBMC, bone marrow) and aneuploid datasets (HCT116, SNU601, Colo320, pediatric glioblastoma), and also including different chemistries.

(5) Related to the previous comment – WES is being compared to epiAneufinder in tumor samples, whereas scWGS is being compared to epiAneufinder in cancer cell

lines. Could the level of agreement be affected by the nature of these samples? Ideally, both WES and scWGS would be compared to the new algorithm both in cell lines and in primary tumors (separate comparisons, but same type of samples).

As described above, we have seen that WES was actually not a suitable ground truth for CNV calling. We have therefore decided to discontinue using WES for comparison. Instead, we now added to the manuscript several more scATAC-seq datasets with (sc)WGS as reliable ground truth for testing our algorithm (see above).

(6) Lastly, the comparison to DNA-based sequencing approaches (scWGS and WES) are indeed the most important comparisons. Nonetheless, it would be highly valuable if the authors could compare their algorithm to those that infer CNVs from scRNAseq data (inferCNV, CaSpER, HoneyBADGER). While inferring CNVs from scRNAseq data can be challenging, as noted by the authors, it has pro's and con's compared to the scATACseq data, and a head-to-head comparison would be informative.

We very much thank the reviewer for this comment. We have now used 3 different scRNA-seq CNV methods (InferCNV [11], CaSpER [12] and copyKat [13]), to compare their results to the ones from epiAneufinder. We also tested HoneyBADGER, as suggested by the reviewer, but it was not feasible to run it on large (droplet-based) single cell datasets due to very long runtimes. Instead, we included copyKat, a more recent method. Moreover, we have also compared the results from epiAneufinder to the ones from another method called Copy-scAT [14], that calls CNVs from scATAC-seq data.

For these comparisons, we have used 3 different datasets:

- SNU601 cell line where the scRNA-seq, scWGS-seq and scATAC-seq modalities are available but have been measured in different cells [15,16]
- HCT116 cell line with multi-ome data (scCAT-seq) [1] (scRNA-seq and scATAC-seq measured in the same cell); scWGS can be used as ground truth [2]
- COLO320 cell line with multi-ome data (10x) (scRNA-seq and scATAC-seq measured in the same cell)[3] ; bulk WGS can be used as ground truth [4].

We evaluated the correlation between the genome-wide results of each RNA method compared to the scWGS or WGS results using pseudo bulk aggregates. For this, the gene-based CNV results of the RNA methods were mapped to the genomic bins of the (sc)WGS and epiAneufinder results. Only bins where results were available for all methods were included in the comparison, i.e. a far smaller part of the genome is evaluated compared to using only scATAC-seq data. A disadvantage of the scRNA-seq methods is that they all require normalizing the sample by a “normal” (euploid) scRNA-seq reference dataset of the same or similar cell type, which is especially difficult to obtain for cell lines (we used as a reference: for SNU601: control samples

from gastric cancer study [17]; for HCT116: non-tumor tissue samples from colorectal cancer patients [18]; for COLO320: gut cell atlas [19]).

For the SNU601 cell line, the correlation between the CNVs obtained from the scWGS data and the CNV results obtained by the different methods/modalities was overall medium to high (see Figure below). The ATAC-based methods (EpiAneufinder & Copy-scAT) performed better than RNA-based methods. The best RNA method, InferCNV, reached a correlation of 0.61, compared to EpiAneufinder (the best ATAC method) with a correlation of 0.86.

All RNA methods performed worse on the other two datasets, the HCT116 cell line and the Colo320 cell line (see Figure below, ranges between 0.20 and 0.42). This could be due to overall poor performance of the RNA-seq methods, poor choice of reference data (as discussed above), or differences in data quality. Finally, the HCT116 data set was not generated using a 10x protocol, but Copy-scAT works on 10x data as default. Therefore, Copy-scAT was not included in the method comparison for this data set.

Reviewer response Figure 5: genome-wide correlation of the pseudo-bulk CNV results between the different methods, for three cell lines: SNU601, HCT116, and Colo320. Only bins that are included in all methods are considered in the comparison.

Finally, we quantified more carefully the results of epiAneufinder and Copy-scAT for the SNU601 cell line, as they both perform CNV calls on scATAC-seq data. In contrast to the comparison with the RNA methods above, a far larger part of the genome could be included here. Copy-scAT, per default, only provides chromosome-arm resolution copy numbers, visible in the very coarse-grain profile in the line plot. Nevertheless, the estimated CNVs between Copy-scAT and epiAneufinder agree overall well (correlation of 0.76). However, epiAneufinder correlates better than Copy-scAT with the WGS results (0.86 vs 0.74).

Overall, the comparison of epiAneufinder to methods for calling CNVs from scRNA-seq data as well as one method for calling CNVs from scATAC-seq data shows that epiAneufinder has improved performance compared to both. We have added these results in the manuscript in a new section, called “epiAneufinder outperforms other single-cell CNA calling methods” (line 216), and in the main Figure 3 and SI Figure 8.

Reviewer #2 (Remarks to the Author):

epiAneufinder is a tool to detect CNVs from scATAC-seq data. It uses a binary segmentation algorithm to locate CNV breakpoints for each cell by calculating the AD distances between read distributions of neighbouring bins/segments and assigning a putative breakpoint to the genomic position with max AD. Breakpoints are pruned if they show an AD distance lower than the genome-wide mean and copy number is calculated for each bin as “loss”, “gain” or “normal” depending on deviation from this mean. The authors apply the tool to characterize an adenocarcinoma cell line (SNU601), and two basal cell carcinoma patient samples (SU008 and SU006) in order to characterize the karyotype and clone diversity in these published datasets. The paper is well written and clear, however, I believe substantial benchmarking is still required (critical to actually assess performance), and there is a lack of interpretation and biological meaning assigned to the genetic subclones described in the study.

Major comments

(1) By comparing against genome-wide mean coverage, the tool assumes a diploid/normal baseline karyotype for the sample and does not consider for instance whole genome doublings, which are indeed common to cancers. This should be discussed.

We thank the reviewer for raising up this very important point. Indeed, we cannot know if the baseline of any sample is diploid, or if the sample for example has a genome duplication. For this reason, we had called the baseline state “normal” instead of “diploid” (“normal” referring to the sample baseline). We have now discussed this limitation with more detail in the manuscript (line 132): *“Precise quantification of copy numbers beyond “gains” and “losses” is not possible due to the sparsity of scATAC-seq data; and full genome duplications and deletions cannot be identified, because they change the genome-wide mean openness value.”*.

(2) To define CNV breakpoints a binary segmentation method iteratively calculates AD for each segment until a “stopping criteria” is reached. This is by default set to seven (line 117). The value seems arbitrary. Why did the authors select this value as default? Is the same value applied to every chromosome? If so, how does it account for chromosomes of differing lengths, and do shorter chromosomes suffer too many breakpoints? How does changing the value alter CNV predictions?

To study how much the stopping criteria changes the CNV predictions, we did run the algorithm on the SNU601 dataset (the one for which we have the best ground truth reference) with stopping criteria ranging from 3 to 31 break points per chromosome, and

we compared the results. The CNV profiles are highly correlated no matter what stopping criteria was used:

a)

b)

Reviewer response Figure 6: a) karyograms for the same dataset (SNU601) using 3, 7, 15 (highlighted) and 31 break points. The cells are ordered in the same order in all plots. b) correlation of pseudo bulk CNV profiles for the following number of breakpoints: 3, 7, 15 and 31.

We now decided to keep 15 breakpoints as a default stopping criteria, as it does not increase the runtime dramatically compared to 31 breakpoints (running time for 15 breakpoints ~22 hours, while for 31 breakpoints ~36 hours, see SI Table1) and could potentially provide higher resolution per chromosome compared to the lower stopping criteria. The stopping limit can be changed by the user.

These results are presented in the manuscript: “*Varying the total number of breakpoints called per chromosome did not substantially change the CNA results*” (line 160) and SI Figure 2 and SI Table 1. Additionally we would like to mention that by running the algorithm with 15 breakpoints as a default we now obtain slightly different clustering results and cluster numbering. This is because the stopping limit used to run the algorithm has an impact on the level of noise in the calls.

(3) Additionally, after initial segmentation, the breakpoints are “pruned” if the AD distance is lower than the genome-wide mean (line 119). This seems to imply the authors assume a mostly ‘normal’ or diploid background for each input cell, which may indeed be erroneous and lead to false negatives. e.g. whole genome doubling events will escape detection, which are common in cancers. This should be clearly discussed. We apologize about the fact that this point was not clear in the manuscript.

The assumption behind “pruning” the breakpoints is that the number of breakpoints called by the algorithm per chromosome is larger than the true number of breakpoints. This happens because there are always 15 breakpoints called per chromosome (the stopping criteria discussed in point (2)). When there are less than 15 true breakpoints in a chromosome, the remaining breakpoints called by the algorithm but not existing in the data have a very low value of the Anderson Darling distance. These are the ones that we remove in this step. Moreover, from the analysis of the total number of breakpoints performed above in the SNU601 cell line, we see that the results of the pruning are robust when using very different numbers of breakpoints. The “pruning” of the breakpoints has therefore nothing to do with the assumption that there is a mostly normal or diploid background for each input cell. We apologize about the fact that this was not clear in the manuscript. We have now described this on the manuscript and have noted that for a sample where a lot of breakpoints are suspected per chromosome, the stopping criteria should be increased (line 120: “*After all the breakpoints have been identified genome-wide, epiAneufinder prunes out breakpoints with an AD distance lower than the genome-wide mean, to remove low AD distance breakpoints. This pruning procedure assumes that less than 15 breakpoints are present per chromosome. In situations where more breakpoints are expected, the parameter for the upper number of breakpoints can be modified by the user.*”).

(4) It seems this breakpoint approach is prone to error, which is likely due to the genome being ‘artificially’ binned prior calculating AD – this is especially clear in Fig S1 where several “loss” bins have a read density similar to the “normal” distribution (hidden behind the normal/gain areas of s1a and visible in s1c). Can a breakpoint refinement approach (e.g. sliding window, and/or pseudobulk clustering) be implemented to better define a more real CNV location?

What is the size distribution of CNVs identified? How balanced are the "gains" to "losses"? what is the limit of detection?

Our algorithm looks for changes in read distribution along the chromosome, considering the vector of (normalized) number of reads per bin. Indeed, it can happen that a breakpoint happens in the middle of a bin. However, if there is a true breakpoint, and the distribution of the reads at the left and right of this bin are different, the breakpoint will be called by the algorithm either to the left or to the right of that bin. This should not lead to incorrect breakpoints, but indeed it could lead to slight deviations in the position of the called breakpoint compared to the real start of the CNV.

To study this, and related to the second question, we have investigated the size distributions of the CNVs identified for several datasets:

Reviewer response Figure 7: size distribution (in units of bins, each bin = 100kb) for the discovered gains, losses and normal part of the genome, for the SNU601 cell line.

The minimum CNV size that we have discovered is 100kb long (1 bin). This value is of course very much dataset dependent. What it tells us though, is that the CNVs that we identify are at the limit of the resolution of the algorithm, as the minimum size is the size of one of the artificial bins. This means that, when the changes in read distributions are statistically significant, a copy number segment for an individual bin can be called by the algorithm.

Implementing a sliding window approach would potentially allow to find the exact position of a breakpoint more accurately (despite the fact that the resolution will always be limited by the extreme coverage sparsity inherent to scATAC-seq data). However, running the algorithm by default using a sliding window would be prohibitive in terms of run-time.

We have introduced these results in the manuscript: “*The smallest CNV identified was 100 kb (1 bin), the largest gain was 266 700 kb and the largest loss was 361900 kb*” (line 159) and on SI Figure 2.

(5) The heatmap and clone composition shown in e.g. Fig 2/S2 contains considerable mixing of karyotypes (e.g. mentioned on line 156). How do the authors interpret the subclone structure? It is reasonable to think the same region would delete multiple times in each subclone independently? Or are these false calls? The authors should discuss this.

The evolution of subclonal structures in karyotypes is an ongoing area of cancer research. Moreover, in the case of cell lines, their evolutionary history tends to be very complex. There are several papers that have studied these complicated phylogenies using single-cell whole genome sequencing data (for example [15]). To interpret the subclone structure from the karyotype of a given sample, we would need to develop a phylogeny algorithm that uses the scATAC-seq copy number variations to reconstruct

the evolution of a given sample. Despite it being very interesting, this is unfortunately out of the scope of this paper.

(6) There is little meaning or biological analysis of the clones defined. Why are some cells grouped into one clone when they show very diverse karyotypes? Are there specific features driving this, or any genes/pathways implicated? Overall, an interpretation of the clone structure and biological relevance is missing for all samples.

First, the cells are grouped together based on their genome-wide karyotype profiles. Basically, every cell is represented as a vector of bins along the genome, and every bin is assigned to the value 1, 2, 3 depending on their copy number call. Then, the Ward's method is used to cluster these genome-wide profiles based on the euclidean distance between vectors. Therefore, by definition, cells grouped together into one cluster have a higher karyotype similarity between them than cells grouped in separate clusters. We do this formally by grouping the cells into clusters using Ward's hierarchical clustering, as clustering the cells together visually based on their genome-wide profile would be very much error prone.

Of course, the population of cells can be split into more or less clusters by using a higher or lower cutoff distance to cut the dendrogram. For visual representation we have used a distance that separated main differences but didn't lead to too many clones. A lower cutoff distance would lead to many more clusters, which would be more homogeneous. No genomic annotation (genes/pathways) was used for the clustering analysis.

For the biological interpretation of the discovered clones, we have used the BCC sample, as for the cell lines their evolutionary history in the lab is very complex (see point above). We have explored the identity of the cells by identifying different marker genes at the proximity of differentially open regions, which are up or down regulated per clone.

First, by exploring the fibroblast clones we have concluded that instead of fibroblasts these cells are cancer associated fibroblasts (CAFs). It has previously been described that CAFs can harbor CNVs in their genome [20]. To conclude that these cells are CAFs, we have shown in both tumor samples that:

1. These cells are not cancer cells as they don't have cancer markers expressed (KRT5, KRT15, TP63, TERT)
2. These cells have fibroblast and cancer associated fibroblast markers expressed (COL1A2, LUM, FAP, VEGFC, ANGPT1, PDGFRB, IL6, etc).

The openness of these marker genes can be visualized via UMAPs and violin plots, per cell type and also per karyotype clone:

Reviewer response Figure 8: marker gene openness per cell type and karyotype, for the sample 006. This figure is now included in SI Figure 10.

It can be seen from the figure that the fibroblast cells have active markers of CAFs and not for tumor.

We also compared the differentially open peaks between karyotypes (top 200 differential regions) and performed GO term enrichment (using the GREAT analysis tool [21]). In sample 006 for example, the peaks which are differentially open in the CAF karyotype with CNVs (karyotype 3.1) compared to the one without CNVs (karyotype 1.2) show “positive regulation of cell migration” and “positive regulation of cell motility”, “regulation of cell migration” or “regulation of endothelial cell migration”, which may indicate genes upregulated in the endothelial to mesenchymal transition. For the cancer karyotype 3.2.1 for example, we found “regulation of MAPK cascade” or “positive regulation of MAPK cascade” as enriched GO terms when compared to the other cancer karyotypes, which may hint to a dysregulation of MAPK signaling. Overall, these results suggest that there could be different biological pathways active in the different karyotypes. These differences could be followed up with subsequent molecular studies, which are out of the scope of this manuscript.

a)

b)

Reviewer response Figure 9: a) GO term enrichment from the GREAT analysis tool for the top differentially accessible peaks in the CAFs with aneuploidies (karyotype 3.1) and b) for the top differentially accessible peaks in one of the cancer clusters that show aneuploidies (karyotype 3.2.1).

Finally, a paper exploring four BCC samples using scRNA-seq was published recently [22]. In it, the authors use scRNA-seq data to look at the karyotypes of individual cells in four samples. By doing so, they discover a great amount of heterogeneity in copy number profiles, with some samples being nearly disomic while others contain a lot of CNVs, which are also heterogeneous between cells of the same sample.

We have added some of these results (also for sample SU008) in the SI figures 10 and 11, and in the main text (paragraph starting in line 322).

(7) Looking at the pseudo-bulk CNV values of Fig 2c, it is hard to believe they are “highly correlated” with a value of 0.86. There are very clear differences between the karyotypes identified in each sample, and this feels ignored. As briefly mentioned in text (lines 180-6), several areas are not called consistently between the modalities (including e.g. losses on chr9q often ‘missed’ by epiAneufinder). Can you show the correlation value for every bin? Why do some chromosomes or regions appear to ‘suffer’ more than others?

Unfortunately, we cannot calculate a correlation per bin. We have summarized the results of the single cells as a pseudobulk result, and we have only one vector per modality that contains the pseudo-bulk genome-wide copy number profile. We show the correlation per chromosome or along the whole genome between these two vectors. The correlation result shown on the manuscript is the genome-wide correlation between these two vectors.

Following the suggestion from the reviewer, however, and to better compare between the epiAneufinder results and the whole genome sequencing results, we have now computed, instead of correlations, the precision, recall and F1 scores. In this way we compare the two pseudo bulk results, bin by bin. We convert the signal into a

classification using an upper threshold for calling a bin as gained, and a lower threshold for calling a bin as lost in both signals. Then, we calculate the confusion matrix for every threshold (true negatives, false negatives, true positives, false positives) and from it we can calculate precision, recall and F1 per state.

We obtain excellent results for the SNU601 dataset that we had already presented in our previous version of the manuscript:

	precision	recall	F1
loss	0.939979123173278	0.82350251486054	0.877894223738728
normal	0.871109930961232	0.99081848384174	0.927116009608591
gain	0.992428153501979	0.737939859245042	0.846469983854396

Reviewer response Figure 10: precision, recall and F1 values for every one of the three states, when comparing the scATAC-seq results to the scWGS results for the SNU601 cell line.

We also obtained very high values of the precision, recall and F1 for the new datasets that we have included in the manuscript. These new results are included in the manuscript in SI Figures 3, 4, 5 and 6.

(8) How does gene activity change the CNV calling? Can the authors test how CNV gains/losses correlate with number of highly/lowly expressed genes per segment?

We thank the reviewer for this very interesting comment. To study this, we have calculated the aggregated gene activity per bin per cell (defined as the sum of the gene activity of all the genes overlapping the bin) and calculated the mean (aggregated) gene activity per CNV status (loss, normal, gain) (for the BCC sample). As expected, the gained bins have a higher gene activity than the lost bins:

Reviewer response Figure 11: mean gene activity in bins identified as lost, as normal and as gained for the SU006 sample.

This is expected as gene expression can also be used as a proxy for copy number variations, and gene activity is a proxy of gene expression.

We also calculated correlation of the aggregated gene activity counts with the CNV calling results per cell, to subsequently calculate the mean correlation across all cells. Genome-wide, the correlation between the gene activity value per bin to the copy number variation per bin is nearly negligible, at 0.12 for both BCC samples.

(9) This validation approach fails to consider the accuracy of CNV calls in each individual cell and comparing to the scWGS data as the “ground truth” is confounded by the clone diversity. I do appreciate having a perfect ‘truth set’ is nearly impossible here, but I think more benchmarking is needed and comparison of ‘true’ CNVs called per cell is missing. Comparing outputs against other CNV tools can help with this.

Can the authors identify the most-likely shared clone between the ATAC and scWGA datasets of Fig2 and compare the CNV recall rates between these (pseudo-bulk’ed) cells?

Is there a scRNA-seq + scATAC-seq dataset that can be included? This might be useful for performing the benchmarking. Despite CNV-calling from scRNA-seq being rather imperfect, this could give a better head-to-head comparison of precision/recall per cell.

We thank the reviewer for these comments which add strength to the results presented in our manuscript. To address these concerns, we have performed a series of validations:

First, we have obtained further datasets where the scATAC-seq profiles can be compared to the ones from whole genome sequencing (WGS) or single-cell whole genome sequencing (scWGS). We have included cell lines, but also primary samples where we have an orthogonal confirmation of the copy number variations:

- HCT116 cell line ([1] and [2] for scATAC-seq and scWGS respectively)
- Colo320 cell line ([3] and [4] for scATAC-seq and WGS respectively)
- Primary and relapse patient pediatric glioblastoma dataset ([5])

To compare the results between scATAC-seq and whole genome sequencing we now compute the precision, recall and F1 of the calls using the (sc)WGS as a ground truth. We do that to better compare the two results, instead of using correlations (see please answer 7 above).

In all the analyzed datasets, comparing the epiAneufinder CNVs to the (sc)WGS led to very high values of the precision, recall and F1. This can be visualized with the karyotypes for all datasets, as well as the genome-wide pseudo bulk profiles:

HCT116 cell line:

	precision	recall	F1
loss	0.623946037099494	0.501355013550135	0.555972952667167
normal	0.983336096833029	0.988169624260602	0.985746935383337
gain	0.95875	0.975206611570248	0.966908288685786

Reviewer response Figure 1: HCT116 cell line: scATAC-seq karyotype (top), scWGS karyotype (middle), and pseudo bulk comparison (bottom). The table shows the best precision, recall and F1. Note that the pseudo bulk for the sc genome sequencing (GS) is calculated from only 22 cells, hence the jagged single cell genome sequencing profile, especially seen in the small sharp losses which are mostly due to noise in the coverage.

Colo320 cell line:

	precision	recall	F1
loss	NA	NA	NA
normal	0.933056278790306	0.922205176424011	0.927598994499614
gain	0.649810648251281	0.685707569346497	0.667276678485646

Reviewer response Figure 2: Colo320 cell line: scATAC-seq karyotype (top) and pseudo bulk comparison to WGS data (bottom), for replicate 1. The table shows the best precision, recall and F1 (no losses identified). The karyotype for all replicates can be found in SI Figure 5.

Pediatric glioblastoma:

	precision	recall	F1
loss	0.774386594853381	0.889347079037801	0.827895073576456
normal	0.927005851433105	0.925033400959968	0.926018575851393
gain	0.723703344643723	0.694580134915097	0.708842729970326

Reviewer response Figure 3: Primary patient glioblastoma dataset: scATAC-seq karyotype (top) and pseudo bulk comparison (bottom) to WGS data, for replicate 2937. The table shows the best precision, recall and F1. All replicates can be found in SI Figure 6.

As for the comparison between shared clones between the scWGS and the scATAC-seq (instead of comparing all cells to all cells), this is much more evident in the case of the HCT116 sample. For that sample, we have selected the cells that share karyotype between modalities (the top cells in both modalities) and we have calculated the F1 score per state between the two. When doing that, we obtain 0.99 and 0.97 for the F1 score for the states normal and gain, respectively, with nearly perfect precision and recall for these two states (precision=0.99,0.99 and recall=0.99,0.96 for the normal

and gain states, respectively). There were only residual losses identified in this cluster of cells.

These results have been added to the section “epiAneufinder CNAs are concordant with the ones obtained from (sc)WGS” (line 147) and to the SI Figures 4, 5 and 6.

As a side note: the high throughput 10x scWGS kit (called single-cell CNV method) is not sold any more by the 10x company. Because of this, there are not a lot of high-throughput new datasets that quantify CNVs using single-cell whole genome sequencing, as they require an in-house protocol.

As suggested by the reviewer, we have also compared the results from epiAneufinder to the results from other CNV callers. To do that, we have used 3 different scRNA-seq CNV methods (InferCNV [11], CaSpER [12] and copyKat [13]), to compare their results to the ones from epiAneufinder. Moreover, we have also compared the results from epiAneufinder to the ones from another method called Copy-scAT [14], that calls CNVs from scATAC-seq data.

For these comparisons, we have used 3 different datasets, some of these datasets are also multi-ome datasets:

- SNU601 cell line where the scRNA-seq, scWGS-seq and scATAC-seq modalities are available but have been measured in different cells [15,16]
- HCT116 cell line with multi-ome data (scCAT-seq) [1] (scRNA-seq and scATAC-seq measured in the same cell); scWGS can be used as ground truth [2]
- COLO320 cell line with multi-ome data (10x) (scRNA-seq and scATAC-seq measured in the same cell)[3] ; bulk WGS can be used as ground truth [4].

We evaluated the correlation between the genome-wide results of each RNA method compared to the scWGS or WGS results using pseudo bulk aggregates. For this, the gene-based CNV results of the RNA methods were mapped to the genomic bins of the (sc)WGS and epiAneufinder results. Only bins where results were available for all methods were included in the comparison, i.e. a far smaller part of the genome is evaluated compared to using only scATAC-seq data. A disadvantage of the scRNA-seq methods is that they all require normalizing the sample by a “normal” (euploid) scRNA-seq reference dataset of the same or similar cell type, which is especially difficult to obtain for cell lines (we used as a reference: for SNU601: control samples from gastric cancer study [17]; for HCT116: non-tumor tissue samples from colorectal cancer patients [18]; for COLO320: gut cell atlas [19]).

For the SNU601 cell line, the correlation between the CNVs obtained from the scWGS data and the CNV results obtained by the different methods/modalities was overall medium to high (see Figure below). The ATAC-based methods (EpiAneufinder & Copy-scAT) performed better than RNA-based methods. The best RNA method, InferCNV, reached a correlation of 0.61, compared to EpiAneufinder (the best ATAC method) with a correlation of 0.86.

All RNA methods performed worse on the other two datasets, the HCT116 cell line and the Colo320 cell line (see Figure below, ranges between 0.20 and 0.42). This could be due to overall poor performance of the RNA-seq methods, poor choice of reference data (as discussed above), or differences in data quality. Finally, the HCT116 data set was not generated using a 10x protocol, but Copy-scAT works on 10x data as default. Therefore, Copy-scAT was not included in the method comparison for this data set.

Reviewer response Figure 5: genome-wide correlation of the pseudo-bulk CNV results between the different methods, for three cell lines: SNU601, HCT116, and Colo320. Only bins that are included in all methods are considered in the comparison.

Due to data sparsity, comparing each cell separately between RNA and ATAC modalities leads to very noisy estimates, as shown in the figure below for the COLO320 dataset (multi-ome dataset for which we have a WGS ground truth) and predictions from copyKat. While the overall correlation is still positive for most of the cells, only a

small fraction got a correlation of at least 0.4, which was the maximum correlation obtained in the comparison between RNA and ATAC from the pseudo bulk comparisons.

Reviewer response Figure 12: per single-cell, correlation between the RNA and the ATAC copy number variations called in every bin, for the Colo320 cell line.

Finally, we quantified more carefully the results of epiAneufinder and Copy-scAT for the SNU601 cell line, as they both perform CNV calls on scATAC-seq data. In contrast to the comparison with the RNA methods above, a far larger part of the genome could be included here. Copy-scAT, per default, only provides chromosome-arm resolution copy numbers, visible in the very coarse-grain profile in the line plot. Nevertheless, the estimated CNVs between Copy-scAT and epiAneufinder agree overall well (correlation of 0.76). However, epiAneufinder correlates better than Copy-scAT with the WGS results (0.86 vs 0.74).

Overall, the comparison of epiAneufinder to methods for calling CNVs from scRNA-seq data as well as one method for calling CNVs from scATAC-seq data shows that epiAneufinder has improved performance compared to both. We have added these results in the manuscript in a new section, called “epiAneufinder outperforms other single-cell CNA calling methods” (line 216), and in Figure 3 and SI Figure 8.

(10) The poor correlation seen between scATAC and WES (line 213) is a little alarming. Can the authors adjust parameters (e.g. binning) to improve this? Would it be an option to compare the consistency between the most-frequent CNV status/bin of the sc data to the bulk WES data?

We agree with the reviewer that the level of agreement with the WES data is, despite being non-negligible, low.

We have investigated this issue further. For the dataset that we used, we cannot know what is the real CNV landscape of the tumor, as we have no alternative measurement

for it (FISH, karyotyping or other). We have however analyzed the WES data with two widely used CNV callers that have optimized parameters for CNV calling on WES data, namely CNVkit [8] and Control-Freec [9].

We found very different CNV results depending on the method used: for tumor 006, Control-freec identified 68 loss regions and 3 gain regions. The loss regions had a total number of 1,968,809,062 bases (almost 2Gbases) and the gain regions 94,165 bases. For the same tumor, CNVkit identified 56 loss regions for a total of 2.5Gbases.

For tumor 008, Control-freec identified 4 loss regions and 51 gain regions. The loss regions had a total number of 5,157,912 bases (5Megabases) and the gain regions 298,637,231 bases (298Mbases). For the same tumor, CNVkit identified 55 loss regions for a total of 2.5Gbases.

For both tumors, there was some overlap between the losses that were identified by CNVkit and the ones from Control-freec, but overall Control-freec identified more “lost” regions. In addition, CNVkit was unable to identify any regions of gain, while Control-freec identified a few gain regions.

Despite our initial surprise, these results corroborate the ones from a recent benchmarking paper from Gabrielaite et al. [9]. In that article, the authors evaluate the performance of CNV calling methods from WES data and conclude that *“In summary, by reviewing 50 tools for CNV calling, of which 11 were included for a benchmark (CLC Genomics Workbench (WGS and WES), cn.MOPS (WGS and WES), CNVkit (WES), CNVnator (WGS), CODEX2 (WGS), Control-FREEC (WGS), DELLY (WGS), ExomeDepth (WES), GATK gCNV (WES and WGS), Lumpy (WGS), and Manta (WES and WGS)), we conclude that CNV identification from NGS data remains challenging. For the best reliability of CNV calling from NGS data, we observed that even if the tools were developed for WES data or allowed it as input, they did not perform well. We suggest WGS as the only NGS-based option for broad calling of CNVs. Furthermore, low precision in all tools leads us to recommend a hypothesis-based approach for finding causative CNVs by NGS in the clinic, and further validation of these candidates by manual inspection, MLPA or array-based approaches.”*. Our results are therefore in agreement with their conclusions.

After our analysis and comparison with the benchmarking paper cited above, we unfortunately concluded that WES is not the best data modality to use as ground truth for CNV calling. We would like to apologize for not realizing this beforehand, and thank the reviewer again for raising the concerns that led to this realization.

We have therefore removed the comparison to the WES results, which cannot be used as ground truth. Nevertheless, we have decided to keep the scATAC-seq CNV results from the BCC dataset in the paper, without orthogonal ground truth. Based on our

comparisons between epiAneufinder results and the ones from (sc)WGS data, we find it safe to assume that epiAneufinder works robustly for calling CNVs from scATAC-seq data. Considering the extra datasets that we discussed in point (9) above, and considering that we have also included several euploid samples (see answer to reviewer 1, point 3, above), we have now in total seven successful comparisons to (sc)WGS data (with multiple technical and biological replicates), including both euploid (brain, PBMC, bone marrow) and aneuploid datasets (HCT116, SNU601, Colo320, pediatric glioblastoma), and also including different chemistries.

(11) In Fig 3 the “fibroblast” classified cells (clone 1.1) show more genetic abnormalities than some “tumor” cells (clones 1.2, 1.3). Why is that?

This question is related to the biological interpretation of the subclones above (question number (6)). It is difficult for us to know why in this particular sample this is the case. As mentioned above (question (6)), the CNV profile of BCC tumor cells can be extremely varied (see for example [22]), both between samples and inside of the same sample. What we can confirm though, is that the “fibroblast” classified cells show markers of CAFs while the “tumor” cells show markers of cancer (see Figure for the reviewers number 8).

(12) Since clone composition appears extremely diverse and the clones are not fully characterized, I have trouble with some of the claims, for example on lines 223-227. The authors say that SU006 contains 8 karyotype clones, but the clone borders do not appear to define karyotypes (see above comment). And indeed clone 1.2 appears karyotypically normal, but is still being called “tumor”. Moreover, clone 3.1 in SU008 contains a huge gain on chr6 that is shared with tumor cells but is assigned as fibroblast – how do authors interpret these inconsistencies and what is the biological reasoning behind this?

As mentioned in the answer above (11), BCC are known to have a very varied CNV profile. In the already cited paper [22], 4 BCC samples were analyzed and every one of them presented a different CNV profile. Moreover, there was heterogeneity present at the level of CNVs inside of the same sample. It is therefore not surprising to find cancer cells which are diploid, or to find fibroblasts (CAFs) that are non-diploid.

What we have done in the new version of our manuscript, is to use the scATAC-seq data to explore more the characteristics of every individual cell type / karyotype. Our findings show that the cells labeled as “fibroblasts” are actually cancer associated fibroblasts, while the cancer cells all have markers of cancer even when they display a normal karyotype (see Reviewer response Figure 8). We have added these results in the new version of the manuscript (paragraph starting in line 322, and SI Figures 10 and 11). Moreover, we have modified the text to not state that we identify karyotype clones.

We instead refer to them with the more technical term of “karyotype clusters”, because they are the clustering result of the Ward’s clustering method.

(13) As I understand CNV calls are very much dependent on depth of coverage. This is true for all calling algorithms, even on bulk data. Thus I have reservations about the down-sampling experiment results. Is there surprisingly high performance at the 20% subset- is this because they are only comparing bins that are shared with 100% set? How many bins were lost during the down-sampling? Why is the performance not reducing at expected rates? How many clones are captured with each subsampling, how many are lost?

We thank the reviewer for this comment, that has helped us clarify this issue in the manuscript. As stated in our manuscript, upon every downsampling experiment we filter the cells that do not pass the algorithm quality control (i.e. not have more than 20,000 reads per cell). SI Table 6 reports the number of cells that passed quality control per downsampling situation. In the cells that pass quality control, the correlations between the fully covered dataset and the downsampled one are very high. Considering cells that do not pass quality control would probably lower that correlation. However, we would not advise using these cells for calling CNVs as they have not passed the necessary quality control filters. We have added a sentence in the results section to make this filtering more clear: “*For every downsampled dataset, cells and bins that did not comply with the quality controls of epiAneufinder were removed (SI Table 6)*” (line 255).

We have also added to the SI Table 6 the number of bins that are retained per subsampling, as bins are filtered if they are not covered in more than 85% of the cells. Also, now that we have changed the parameters for the number of breakpoints (as suggested in point (2) above) we observe a far greater impact of the coverage on our results, more in line with your expectations. This is because with more resolution, we also get more noise.

Finally, we have calculated another downsampling datapoint, at 10% of coverage. At that point the correlations are becoming much lower, indicating that indeed when so few cells are left with sufficient coverage to calculate CNVs, the results from the fully covered dataset cannot be recovered any more. This 10% subsampling point is now added to the main figure 3:

Reviewer response Figure 13: percent of bins which are not detected in the original CNV state any more upon downsampling. And similarity between the downsampled dataset and the full dataset.

(14) The authors claim that after down-sampling the overall copy number profiles are recovered across the genome (line 300) but these data are not shown, and hard to believe.

We are sorry for this omission. The resulting karyotypes for every downsampling experiment can now be found in the new SI figure 9. The cells are ordered in the same way as for the fully covered dataset.

Minor comments

CNVs are very broadly defined as ≥ 2 are “gains”, can better resolution of copy number values be achieved?

Unfortunately, with scATAC-seq data it is not possible to find this level of resolution. We comment that now in the text, line 132: *“Precise quantification of copy numbers beyond “gains” and “losses” is not possible due to the sparsity of scATAC-seq data; and full genome duplications and deletions cannot be identified, because they change the genome-wide mean openness value.”*

precision and recall should be plotted as curves

We have now implemented this suggestion, the new figures are in SI Figure 9.

Fig1A – is this real single-cell ATAC-seq data? For which cell? How representative is this to expected coverage for scATAC-seq? I ask, because in the manuscript it says the median coverage is $\sim 75k$ fragments/cell, thus with the genome binned into 100kb segments I would expect $\sim 2-3$ reads/window, but instead ~ 10 is shown for “normal”.

This is indeed a single-cell, for the SNU601 dataset (cell-ACTGCAATCGGGTCCA-1). This cell has 26557 bins (after removing blacklisted regions and low coverage bins). The total number of fragments for this cell is 160592, giving an average of ~ 6 fragments

per window. A fragment is basically a read pair, so if we multiply the fragments by 2 then we have an average ~12 reads per window.

Fig1B – the CDF panels on the far right are not explained and unclear what they are showing

These panels have now been removed.

Fig S1 appears to be a replicate of Fig 1 – would be nice to see also other representative cell(s) with differing karyotypes

We have now added several cells to Figure S1 with different karyotypes and from different datasets.

Fig S2 (etc) - The clones should also be indicated, to provide reference with the dendrogram

This has been added.

Fig s3b - I found this file unmanageable, and am not sure what it is showing. Is there option to reduce size and improve readability.

We are sorry about that, this is the standard output of the program used for calling CNVs from single-cell whole genome sequencing data. We also show the same results in the “epiAneufinder” format in SI Figure 3.

Fig 2c – axis is not labelled.

The x-axis has now been labeled.

Fig s8 – the clone colors are not consistent between a and b – it would also be easier to read if the colors were darker and clone-themed (e.g. 3.1-3.4: shades of green vs 1.1-1.3: shades of blue)

We have changed the colors to be the same between a and b and we have also changed the color scale to make it easier to interpret.

The sentence on lines 230-232 seems to overstate claims. There tumor cells were not identified by their copy number profiles, as I believe that is why a classifier was used. Also there appears to be little relationship between the karyotypes of “tumor cell” clones, and very little concordance between the UMAP clusters and clones, so I do not believe this sentence is supported by data.

We have now rephrased this sentence to “*These results highlight the power of epiAneufinder to identify novel sources of heterogeneity in the population of cells, based on their copy number profiles.*” (line 350).

Reviewer #3 (Remarks to the Author):

The authors describe epiAneufinder, a tool that enables the detection and profiling of copy number alterations from single-cell ATAC-seq datasets. This tool, similar to methods that leverage scRNA data for CNA calling, seeks to gain additional information out of tumor samples other than the primary modality. The tool fills a very important need – often samples have epithelial cell types that are difficult to distinguish between tumor-adjacent epithelial cells or cancer cells, and a measure of aneuploidy is a powerful way to make this determination. Therefore the impact of the work is high and I expect there will be a great deal of use by the method. While the results are somewhat noisy – they are clean enough to aid in meaningful biological interpretation and the tool mars a nice step forward an analytical methods for ATAC data.

(1) Note: CNV is typically used to encompass germline variation within populations as opposed to CNA (copy number alteration) which is more general and applicable to cancer, where this tool is most appropriate. I don't think it matters really, since papers often use them interchangeably, so it is just a suggestion.

We thank the reviewer for this comment, we have implemented it on the manuscript.

(2) A notable strength is that the tool is very easy to implement. We did not have any challenge running it according to the documentation on our own datasets on the first try. This is in contrary to many other tools that get released.

We thank the reviewer for this comment.

(3) The results appear to be fairly noisy – we also observed this on our datasets; however, we were able to extract general trends – ie cells from a certain cluster are more frequently showing a certain aneuploid profile, thus allowing us to identify that cluster as cancer vs tumor-adjacent epithelial. A useful function would be the addition of a metric for “estimated probability of aneuploidy”. This would be very useful, as one could cross-reference with identified clusters based on ATAC signal and perform this discrimination and it would not rely on having really high coverage ATAC data to get granular copy number calls. (not a requirement for publication, just a suggestion of something that would be super useful in many cases) Overall, the noisiness is much lower than I would expect from ATAC-based CNV calling, so the tool appears to work quite well – this suggestion is more to accommodate noisier datasets as well.

We thank the reviewer for this suggestion. We are planning to continue developing the current algorithm to include scRNA-seq data to it (multi-ome data) and including a statistical measure of “estimated probability of aneuploidy” will be done as an upgrade to the method, using the bi-modality information of the multi-ome data.

(4) On the primary patient samples, all cells are ‘tumor’ since the sample is from a tumor – the cells labeled as such are cancer cells / neoplastic cells within the tumor.

We thank the reviewer for this comment, we have now labeled the cells accordingly.

(5) Is it possible to normalize against a sample of known euploid cells? i.e. a known population of cells with a similar cell type to a suspected neoplastic cell cluster – and then get cleaner results?

We thank the reviewer for this very interesting comment. We have discussed extensively the option of using a normal euploid sample to normalize the data, before performing CNV calling. However, we have reached the conclusion that one of the strengths of our method is that we do not need a reference sample for normalization. Since scATAC-seq data is cell-type dependent, the normalization sample would need to be of the same cell type and ideally from the same sample and generated by the same laboratory. This, for example, would not have been available for any of the samples that we have analyzed for this manuscript, indicating that it would not be very practical. Moreover, an orthogonal technique would have to be used to determine that a sample is indeed euploid and can be used for normalisation (i.e. not using scATAC-seq data).

Instead of normalizing by an euploid sample, what we have done is that we have tested that our algorithm can identify euploid samples, without the use of a reference. To address it, we have now run epiAneufinder in 9 euploid samples from different sources:

-Brain single-cell ATAC-seq (4 replicates) and single-cell multi-ome (3 replicates) datasets [6]

-Bone marrow (1 replicate) and PBMC (1 replicate) scATAC-seq data [7]

In these euploid datasets analyzed, epiAneufinder does not find relevant copy numbers. There are a few non-shared small CNVs found in some cells, which can be most likely attributed to noise, as can be seen in the karyotypes for the normal cells shown below:

a)

b)

c)

d)

e)

f)

g)

h)

i)

Reviewer response Figure 4: a-d) Karyotypes for the scATAC-seq brain samples (top), and pseudo bulk results compared to a baseline of “normal”. e-g) Karyotypes for the ATAC-seq modality from the multi-ome data for brain samples (top), and pseudo bulk results compared to a baseline of “normal”. h) Karyotypes for the scATAC-seq PBMC sample (top), and pseudo bulk results compared to a baseline of “normal”. i) Karyotypes for the scATAC-seq bone marrow sample (top), and pseudo bulk results compared to a baseline of “normal”.

Assuming a ground truth of “normal” and comparing our pseudo bulk scATAC-seq results to it (aggregate of the CNV value per cell in every bin, divided by the number of cells) yields a recall of

Sample	Recall
Satpathy_BoneMarrow	0,97459945
Satpathy_PBMC_rep1	0,98485799
Greenleaf_brain_rep1	0,97002464
Greenleaf_brain_rep2	0,97008819
Greenleaf_brain_rep3	0,9710737
Greenleaf_brain_rep4	0,97130869
multiome_brain_rep1	0,98151908
multiome_brain_rep2	0,98316854

multiome_brain_rep3	0,98261042
------------

Please note that the precision cannot be calculated because we only have true positives and false positives in that dataset. In addition, we have also calculated the variation and the mean square error of these results (mean square error considering a ground truth of “normal”), to show how much deviation there is between our results and the expected “normal” result genome wide. These values are very low, showing that we indeed have not discovered any relevant copy numbers:

Sample	Variation	MSE
bone marrow	0,01251366	0,00067758
PBMC	0,00921701	0,00038248
brain rep1	0,01485453	0,00093185
brain rep2	0,01436002	0,00085472
brain rep3	0,01440061	0,00085159
brain rep4	0,01458655	0,00088512
multiome brain rep1	0,01408168	0,00079308
multiome brain rep2	0,013363	0,00071464
multiome brain rep3	0,01374836	0,00075774

With these results, we show that epiAneufinder recognizes absence of CNVs and correctly identifies euploid samples.

We have added the results on the euploid samples in the manuscript in section “epiAneufinder CNAs are concordant with the ones obtained from (sc)WGS” (line 147) and to the SI Figure 7.

Finally, to test the quality of epiAneufinder results, we have also compared our results to the ones obtained from CNV calling methods from scRNA-seq data. All scRNA-seq CNV calling methods require a reference euploid sample for normalization, and we have realized how difficult it is, in the majority of the cases, to obtain it for each dataset.

For these comparisons of epiAneufinder results to the ones from CNV calling methods using scRNA-seq, we have used 3 different scRNA-seq CNV methods (InferCNV [11], CaSpER [12] and copyKat [13]), to compare their results to the ones from epiAneufinder. Moreover, we have also compared the results from epiAneufinder to the ones from another method called Copy-scAT [14], that calls CNVs from scATAC-seq data.

For these comparisons, we have used 3 different datasets:

- SNU601 cell line where the scRNA-seq, scWGS-seq and scATAC-seq modalities are available but have been measured in different cells [15,16]
- HCT116 cell line with multi-ome data (scCAT-seq) [1] (scRNA-seq and scATAC-seq measured in the same cell); scWGS can be used as ground truth [2]
- COLO320 cell line with multi-ome data (10x) (scRNA-seq and scATAC-seq measured in the same cell)[3] ; bulk WGS can be used as ground truth [4].

We evaluated the correlation between the genome-wide results of each RNA method compared to the scWGS or WGS results using pseudo bulk aggregates. For this, the gene-based CNV results of the RNA methods were mapped to the genomic bins of the (sc)WGS and epiAneufinder results. Only bins where results were available for all methods were included in the comparison, i.e. a far smaller part of the genome is evaluated compared to using only scATAC-seq data. As already discussed, the largest disadvantage of the scRNA-seq methods compared to epiAneufinder is that they all require normalizing the sample by a “normal” (euploid) scRNA-seq reference dataset of the same or similar cell type. This is especially difficult to obtain for cell lines (we used as a reference: for SNU601: control samples from gastric cancer study [17]; for HCT116: non-tumor tissue samples from colorectal cancer patients [18]; for COLO320: gut cell atlas [19]).

For the SNU601 cell line, the correlation between the CNVs obtained from the scWGS data and the CNV results obtained by the different methods/modalities was overall medium to high (see Figure below). The ATAC-based methods (EpiAneufinder & CopyscAT) performed better than RNA-based methods. The best RNA method, InferCNV, reached a correlation of 0.61, compared to EpiAneufinder (the best ATAC method) with a correlation of 0.86.

All RNA methods performed worse on the other two datasets, the HCT116 cell line and the Colo320 cell line (see Figure below, ranges between 0.20 and 0.42). This could be due to overall poor performance of the RNA-seq methods, poor choice of reference data (as discussed above), or differences in data quality. Finally, the HCT116 data set was not generated using a 10x protocol, but Copy-scAT works on 10x data as default. Therefore, Copy-scAT was not included in the method comparison for this data set.

Reviewer response Figure 5: genome-wide correlation of the pseudo-bulk CNV results between the different methods, for three cell lines: SNU601, HCT116, and Colo320. Only bins that are included in all methods are considered in the comparison.

Finally, we quantified more carefully the results of epiAneufinder and Copy-scAT for the SNU601 cell line, as they both perform CNV calls on scATAC-seq data. In contrast to the comparison with the RNA methods above, a far larger part of the genome could be included here. Copy-scAT, per default, only provides chromosome-arm resolution copy numbers, visible in the very coarse-grain profile in the line plot. Nevertheless, the estimated CNVs between Copy-scAT and epiAneufinder agree overall well (correlation of 0.76). However, epiAneufinder correlates better than Copy-scAT with the WGS results (0.86 vs 0.74).

Overall, we have shown that epiAneufinder is able to recognise euploid samples, and it does that without the use of a reference euploid dataset. We have also shown that epiAneufinder outperforms other methods for calling CNVs from scRNA-seq data as well as one method for calling CNVs from scATAC-seq data. Because obtaining matched euploid samples is in general complicated (same cell type is needed, the absence of copy numbers should be verified via an orthogonal measurement technique), we concluded that it is better to do not use euploid samples for running epiAneufinder.

We have added the results presented here in the manuscript in a new section, called “epiAneufinder outperforms other single-cell CNA calling methods” (line 216), and in Figure 3 and SI Figure 8.

(6) Are the authors certain that the cancer cell population – subclone 1 – are actually cancer cells? They could just be non-neoplastic epithelial based on the ATAC analysis. Also there are no immune cells identified which is odd to me...

The fact that no immune cells were identified can be attributed to the FACS sorting strategy used by the authors in that publication [7]: only CD45negative / CD3negative cells were selected. This includes only tumor and stromal cells, not immune cells.

As for the diploid cancer subclone (cluster 1), we have checked the markers present in them and have concluded that they are indeed cancer cells, as they have differential openness at the markers that we have used to identify neoplastic cells (using gene activity):

Reviewer response Figure 8: marker gene openness per cell type and karyotype, for the sample 006. This figure is now included in SI Figure 10.

Based on that, we think that the euploid cancer cells are indeed cancer cells.

(7) In figures S8 and S9 I would expect to see some more consistency in the karyotype vs ATAC clusters. In the ‘tumor’ cluster the karyotypes appear randomly distributed... I would expect that distinct subclones would have some shifts in their epigenetic profiles; but this may not necessarily be the case.

We thank the reviewer for this comment. We were also very surprised about this result. We have tried embedding the cells based not on peaks but based on windows. Our thinking behind this strategy was that when embedding the cells based on peaks, the scATAC-seq results are binarised (a peak can either be present (1) or absent (0)), and

this could be the cause why the CNV information is lost. However, in the embedding based on windows (where the number of reads per window are quantified) the correlation between karyotype and ATAC clusters is also mainly absent. For the window embedding we have chosen exactly the same windows that were used for calling the CNVs. The embeddings are the following:

Reviewer response Figure 14: embedding based on peaks (a) and on windows (b), with identified cell types (left) and karyotype clusters (right).

We see however some enrichment for karyotype 2 at the bottom of the tumor cluster, and some enrichment for karyotype 3 at the top of the tumor cluster. Despite the observed enrichment, the separation is not perfect. This indicates that a joint embedding based on CNV and global chromatin openness could provide more refined identification of subcellular heterogeneity.

We have added these results in the manuscript: “*The different CNA clones present in each tumor could not have been identified based on the embedding results, also when the cells were embedded using the same windows used for CNA calling (SI Fig. 10f-g, 11f-g), emphasizing the relevance of epiAneufinder for discovering new sources of variation in the data.*” (line 337).

REFERENCES:

- [1] Liu, L., Liu, C., Quintero, A. et al. Deconvolution of single-cell multi-omics layers reveals regulatory heterogeneity. *Nat Commun* 10, 470 (2019). <https://doi.org/10.1038/s41467-018-08205-7>
- [2] Cohen-Sharir, Y., McFarland, J.M., Abdusamad, M. et al. Aneuploidy renders cancer cells vulnerable to mitotic checkpoint inhibition. *Nature* 590, 486–491 (2021).
- [3] Hung, K.L., Yost, K.E., Xie, L. et al. ecDNA hubs drive cooperative intermolecular oncogene expression. *Nature* 600, 731–736 (2021).
- [4] Wu S, Turner KM, Nguyen N, et al. Circular ecDNA promotes accessible chromatin and high oncogene expression. *Nature*. 2019 Nov;575(7784):699-703. doi: 10.1038/s41586-019-1763-5. Epub 2019 Nov 20. PMID: 31748743; PMCID: PMC7094777.

- [5] Hoffman, M., Gillmor, A.H., Kunz D.J. et al. Intratumoral Genetic and Functional Heterogeneity in Pediatric Glioblastoma. *Cancer Res* (2019) 79 (9): 2111–2123.
- [6] Trevino AE, Müller F, Andersen J, et al. Chromatin and gene-regulatory dynamics of the developing human cerebral cortex at single-cell resolution. *Cell*. 2021 Sep 16;184(19):5053-5069.e23.
- [7] Satpathy AT, Granja JM, Yost KE, et al. Massively parallel single-cell chromatin landscapes of human immune cell development and intratumoral T cell exhaustion. *Nat Biotechnol*. 2019 Aug;37(8):925-936.
- [8] Talevich E, Shain AH, Botton T, Bastian BC. CNVkit: Genome-Wide Copy Number Detection and Visualization from Targeted DNA Sequencing. *PLoS Comput Biol*. 2016 Apr 21;12(4):e1004873.
- [9] Boeva V, Popova T, Bleakley K, et al. Control-FREEC: a tool for assessing copy number and allelic content using next-generation sequencing data. *Bioinformatics*. 2012 Feb 1;28(3):423-5.
- [10] Gabrielaite M, Torp MH, Rasmussen MS, et al. A Comparison of Tools for Copy-Number Variation Detection in Germline Whole Exome and Whole Genome Sequencing Data. *Cancers*. 2021; 13(24):6283.
- [11] Tickle T, Tirosh I, Georgescu C, Brown M, Haas B (2019). inferCNV of the Trinity CTAT Project.. Klarman Cell Observatory, Broad Institute of MIT and Harvard, Cambridge, MA, USA. <https://github.com/broadinstitute/inferCNV>.
- [12] Serin Harmanci, A., Harmanci, A.O. & Zhou, X. CaSpER identifies and visualizes CNV events by integrative analysis of single-cell or bulk RNA-sequencing data. *Nat Commun* 11, 89 (2020).
- [13] Gao, R., Bai, S., Henderson, Y.C. et al. Delineating copy number and clonal substructure in human tumors from single-cell transcriptomes. *Nat Biotechnol* 39, 599–608 (2021).
- [14] Ana Nikolic et al. ,Copy-scAT: Deconvoluting single-cell chromatin accessibility of genetic subclones in cancer.*Sci. Adv.*7, eabg6045(2021)
- [15] Andor, N., Lau, B.T., Catalanotti, C., et al. Joint single cell DNA-seq and RNA-seq of gastric cancer cell lines reveals rules of in vitro evolution, *NAR Genomics and Bioinformatics*, Volume 2, Issue 2, June 2020.
- [16] Wu CY, Lau BT, Kim HS, et al. Integrative single-cell analysis of allele-specific copy number alterations and chromatin accessibility in cancer. *Nat Biotechnol*. 2021 Oct;39(10):1259-1269.
- [17] Kim, J., Park, C., Kim, K.H. et al. Single-cell analysis of gastric pre-cancerous and cancer lesions reveals cell lineage diversity and intratumoral heterogeneity. *npj Precis. Onc.* 6, 9 (2022).
- [18] Zhang L, Li Z, Skrzypczynska KM, et al. Single-Cell Analyses Inform Mechanisms of Myeloid-Targeted Therapies in Colon Cancer. *Cell*. 2020 Apr 16;181(2):442-459.e29.
- [19] Elmentaite, R., Kumasaka, N., Roberts, K. et al. Cells of the human intestinal tract mapped across space and time. *Nature* 597, 250–255 (2021).
- [20] Yerly L, Pich-Bavastro C, Di Domizio J, et al. Integrated multi-omics reveals cellular and molecular interactions governing the invasive niche of basal cell carcinoma. *Nat Commun*. 2022 Aug 20;13(1):4897.
- [21] Cory Y McLean, Dave Bristor, Michael Hiller, et al. GREAT improves functional interpretation of cis-regulatory regions. *Nat. Biotechnol.* 28(5):495-501, 2010.
- [22] Guerrero-Juarez CF, Lee GH, Liu Y, et al. Single-cell analysis of human basal cell carcinoma reveals novel regulators of tumor growth and the tumor microenvironment. *Sci Adv*. 2022 Jun 10;8(23):eabm7981.

REVIEWERS' COMMENTS

Reviewer #1 (Remarks to the Author):

The authors have addressed my main concerns. The analysis of additional cell lines improves the robustness of the manuscript, and the detailed analysis and comparison of the new tool to RNAseq-based methods is valuable. Also, although I'm not sure that removing WES data altogether is necessary (given that it is still commonly used for CN calling, despite its caveats), I agree that for comparison purposes it cannot represent "ground truth" well, and so the decision to remove these data from the manuscript is very reasonable. Overall, the revisions have improved the manuscript, and I agree with Reviewer #3 that "while the results are somewhat noisy – they are clean enough to aid in meaningful biological interpretation". I have no further comments.

Reviewer #2 (Remarks to the Author):

In their revised manuscript, the authors now include an impressive amount of new data (including several cell lines, primary samples, new datatypes and tools), new analyses and more thorough benchmarking of tool performance, and also describe biological features of the identified subclones in their samples. I believe this has vastly improved the manuscript, and I also thank the authors for responding to every reviewer comment with care.

Reviewer #3 (Remarks to the Author):

The authors thoroughly addressed each of my comments. I have no further suggestions.